# Pharmacologic inhibition of protein phosphatase-2A achieves durable immune-mediated antitumor activity when combined with PD-1 blockade

Winson S. Ho[1], Herui Wang[2], Dominic Maggio[1], John S. Kovach[3], Qi Zhang[2], Qi Song[2,4], Francesco M. Marincola[5], John D. Heiss[1], Mark R. Gilbert[2], Rongze Lu[6,7] & Zhengping Zhuang[1,2]

Mounting evidence suggests that inhibition of protein phosphatase-2A (PP2A), a serine/threonine phosphatase, could enhance anticancer immunity. However, drugs targeting PP2A are not currently available. Here, we report that a PP2A inhibitor, LB-100, when combined with anti-PD-1 (aPD-1) blockade can synergistically elicit a durable immune-mediated antitumor response in a murine CT26 colon cancer model. This effect is T-cell dependent, leading to regression of a significant proportion of tumors. Analysis of tumor lymphocytes demonstrates enhanced effector T-cell and reduced suppressive regulatory T-cell infiltration. Clearance of tumor establishes antigen-specific secondary protective immunity. A synergistic effect of LB-100 and aPD-1 blockade is also observed in B16 melanoma model. In addition, LB-100 activates the mTORC1 signaling pathway resulting in decreased differentiation of naive CD4 cells into regulatory T cells. There is also increased expression of Th1 and decreased expression of Th2 cytokines. These data highlight the translational potential of PP2A inhibition in combination with checkpoint inhibition.

[1] Surgical Neurology Branch, National Institute of Neurological Disorders and Stroke, National Institutes of Health, Bethesda, MD 20892, USA. [2] Neuro-Oncology Branch, National Cancer Institute, National Institutes of Health, Bethesda, MD 20892, USA. [3] Lixte Biotechnology Holdings, Inc., East Setauket, NY 11733, USA. [4] Department of Pathology, Zhongshan Hospital, Fudan University, 200032 Shanghai, People's Republic of China. [5] AbbVie Biotherapeutics, Redwood City, CA 94063, USA. [6] MedImmune, Gaithersburg, MD 20878, USA. [7] Present address: AbbVie Biotherapeutics, Redwood City, CA 94063, USA. These authors contributed equally: Winson S. Ho, Herui Wang. Correspondence and requests for materials should be addressed to R.L. (email: lurongzerose@gmail.com) or to Z.Z. (email: zhengping.zhuang@nih.gov)

Checkpoint molecules, such as programmed dealth-1 (PD-1) and cytotoxic T lymphocyte-associated protein 4 (CTLA-4), are negative regulators of immune function. Tumors often take advantage of these negative feedback mechanisms to create an overall immunosuppressive state and escape immunosurveillance[1]. Monoclonal antibodies blocking PD-1 or CTLA-4 signaling can induce durable long-term immune responses in a subset of patients in a broad range of human cancers. However, a large portion of patients fail to or only partially respond to monotherapy[2], highlighting the fact that multiple redundant mechanisms are involved in mediating the evasion of anticancer immunity[3]. Further progress in immunotherapy requires identification of resistance mechanisms and combination strategies that augment the effect of checkpoint inhibition while avoiding autoimmune side effects.

Protein phosphatase-2A (PP2A) is a ubiquitous serine/threonine phosphatase implicated in diverse cellular processes. In the immune system, Taffs et al. was first to demonstrate the potential role of PP2A as a negative regulator of cytotoxic T-cell effector function. Inhibition of PP2A resulted in enhanced antigen-specific cytotoxicity of lymphocytes[4]. A later study by Parry et al.[5] showed that PP2A mediated the inhibitory signaling of CTLA-4 by dephosphorylating Akt in activated T cells. A recent study by Zhou et al. provided further support for PP2A as a target for immunotherapy. Using an in vivo shRNA screen for immunotherapy targets, *Ppp2r2d*, a regulatory subunit of PP2A, when inhibited was found to be the most potent in enhancing the cytotoxic function of tumor infiltrating lymphocytes (TILs)[6]. Silencing of *Ppp2r2d* using shRNA resulted in increased TILs proliferation and cytokine production. There was also decreased tumor burden and increased survival of mice using adoptive transfer of *Ppp2r2d* silenced OT-1 lymphocytes in a B16-ova melanoma model[7]. In addition, PP2A activity was also found to be elevated in regulatory T cells (Tregs) compared to conventional T cells as a result of endogenous activator transcribed by Foxp3[8]. Treg cell-specific deletion of PP2A resulted in Treg dysfunction and impaired immunosuppressive capability via increased mTORC1 signaling[9]. Furthermore, PP2A inhibition was found to reverse hyperkalemia-induced suppression of TILs in a pharmacologic screen[10]. Taken together, this information suggests that inhibition of PP2A is a promising strategy to enhance anticancer immunity. However, no inhibitors of PP2A are currently clinically available. Established chemical inhibitors, such as okadaic acid and cantharidin, are toxic and have limited clinical utility[11].

LB-100 is a first-in-class small molecule inhibitor of PP2A. In a completed Phase 1 study, LB-100 was shown to be well tolerated in adult patients bearing progressive solid tumors[12]. Multiple xenograft tumor models demonstrated that LB-100 acts as an effective chemo- or radio-sensitizer[13], by inducing aberrant cell cycle progression and mitotic catastrophe[14,15]. However, none evaluated the effects of LB-100 on the immune system because all in vivo studies were performed using immunocompromised mice. Given the mounting evidence that PP2A is a promising target for immunotherapy, we hypothesized that its pharmacologic inhibition using LB-100 could enhance immune activation and synergize with immune checkpoint blockade. To our knowledge, this is the first study demonstrating in a pre-clinical model, that inhibition of PP2A pharmacologically can enhance response to immunotherapy.

## Results

**LB-100 and aPD-1 combination elicit rejection of CT26 tumors**. The pharmacokinetics of LB-100 and its known metabolite endothall were summarized in Supplementary Table 1 and Supplementary Table 2. To test the hypothesis that PP2A inhibition synergizes with aPD-1 therapy in vivo in aPD-1 refractory tumors, we used CT26 tumor, which is a murine colorectal carcinoma with high PD-L1 expression but limited response to aPD-1 therapy[16]. Mice were inoculated with CT26 tumor cells ($0.5 \times 10^6$). After 10–13 days, mice with tumors reaching 50–100 mm$^3$ in size were randomized into four treatment groups: control (PBS), aPD-1, LB-100, and the combination of aPD-1 and LB-100. Treatments were administered every 2 days until survival end point. Tumor growth was assessed every 2 days (Fig. 1a).

The dose of LB-100 chosen in this study deserves specific mention. Earlier study showed that inhibition of PP2A using okadaic acid resulted in a dualistic effect on T-cell function. Low level of inhibition enhanced the antigen-specific cytotoxicity, whereas high level of inhibition resulted in diminished T-cell activity[4]. Our own in vitro experiments (see below) reproduced such a response pattern. We therefore hypothesized that a lower dose of LB-100 should be chosen for its effect on the immune system. While previous pre-clinical studies of LB-100 investigating the effect of PP2A inhibition in tumors used a dose of 1.5 mg kg$^{-1}$ in vivo[13–15,17], we chose a significantly lower dose of 0.16 mg kg$^{-1}$ after a series of pilot dose titration experiments. To confirm target engagement of PP2A at such low LB-100 dose, we measured PP2A activity of isolated CD3+ T cells 4 h after third injections of LB-100. PP2A activity of CD3+ T cells in LB-100-treated mice was inhibited by 37% compared to control (Fig. 1b, $p < 0.01$ by Student's two tailed $t$-test).

In the CT26 model, LB-100 alone did not significantly decrease tumor growth, but did extend median survival (33 vs 21 days, $p = 0.02$ by log-rank test). Additionally, aPD-1 alone had no effect on tumor growth or survival. The combination of LB-100 and aPD-1, however, resulted in striking regression of a significant portion of tumors, with 50% achieving complete regression (CR) for the duration of the study. There was a significant difference in tumor size at day 8 after treatment ($p \leq 0.05$ by one-way ANOVA, Tukey's multiple comparison test) and significant increase in survival ($p < 0.005$ by log-rank test) between the combination and control treatment arms (Fig. 1c, d).

**Effect of LB-100 is dependent on adaptive immunity**. Given PP2A is expressed in tumor cells and previous studies demonstrating the chemo- and radio- sensitizing effect of LB-100 on various tumors, we would like to confirm the observed antitumor effect using this low dosage of LB-100 was a T-cell-mediated process. Using immunodeficient NSG mice that lack B, T, and natural killer (NK) cells, we similarly treated CT26-bearing NSG mice with the same treatment schedule as above. There was no observed difference in tumor growth or survival among treatment groups (Fig. 1e, f). This result demonstrated that the effect of LB-100 given at this dosage is dependent on adaptive immunity. The result largely eliminated the possibility that the observed antitumor effect with aPD-1 was a direct effect on tumor, as neither LB-100 alone or in combination with aPD-1, altered tumor growth or survival in the absence of T cells.

**Effect of combination treatment is dependent on CD8+ T cells**. Next, to further demonstrate CD8+ T cells are the critical T-cell component that mediates the synergistic effect of the LB-100 and aPD-1 combination leading to durable tumor regression, CT26 tumor-bearing mice were subjected to CD8+ T-cell ablation using depleting antibodies prior to and during combination treatment (Fig. 1g). Peripheral CD8+ depletion was confirmed 5 days after treatment by FACS. When depleted of CD8+ T cells, the LB-100 and aPD-1 combination did not elicit tumor rejection (0 vs 72%, $p = 0.0015$ by $\chi^2$ test) (Fig. 1h, i). Mean tumor volume

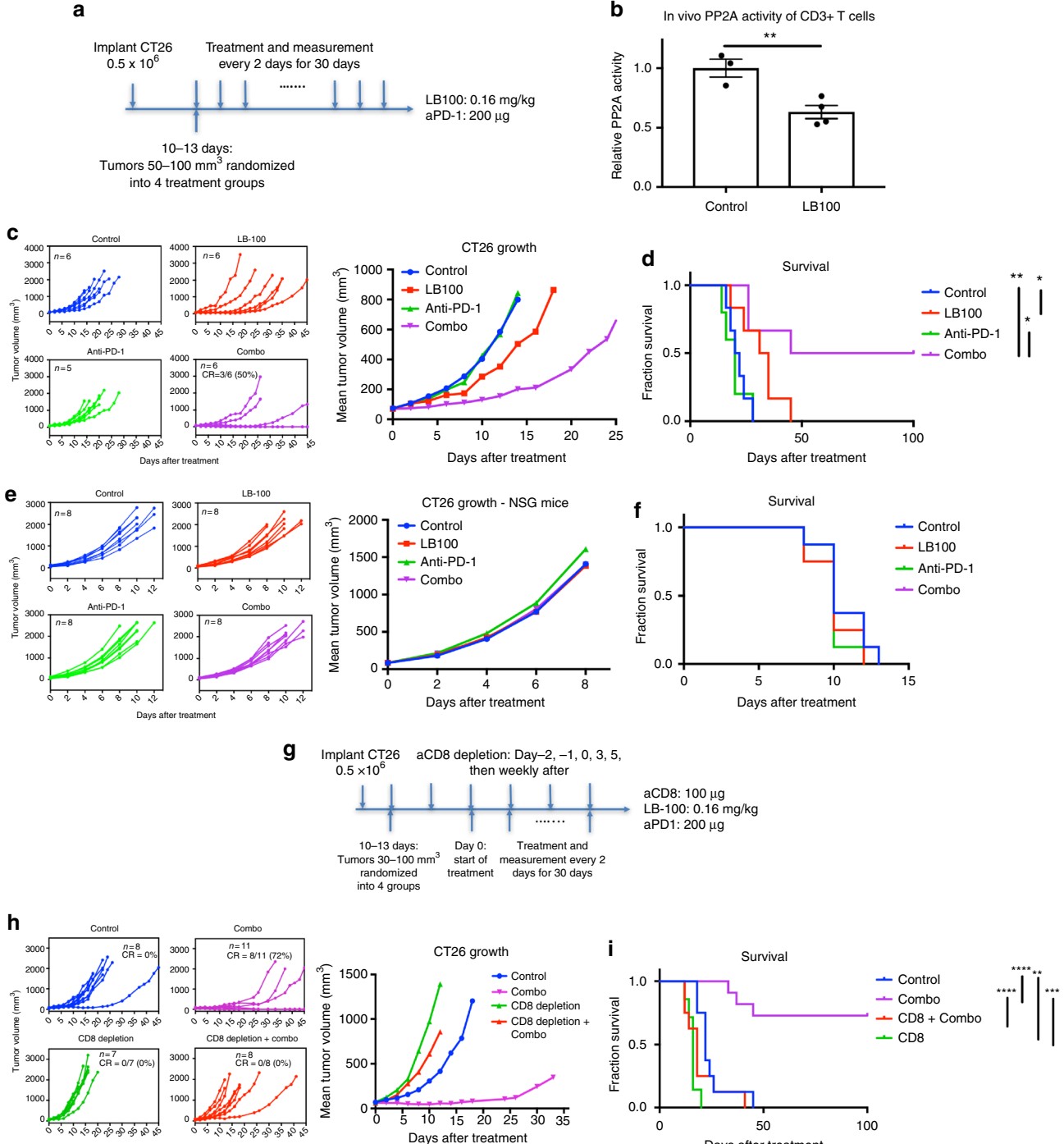

**Fig. 1** PP2A inhibition and PD-1 blockade synergistically elicit tumor rejection in a CD8+ T cell-dependent manner. **a** BALB/c mice were inoculated with $0.5 \times 10^6$ CT26 cells subcutaneously in the right thoracic flank. When tumors reached between 50 and 100 mm³, mice were then randomized to four treatment groups and treated every 2 days until reaching survival end point. **b** BALB/c mice were treated with PBS or LB-100, 0.16 mg kg⁻¹, every other day. 4 h after the third injection, CD3+ T cells were isolated from the spleen. PP2A activity was measured relative to control ($n = 3$ for control and $n = 4$ for LB-100 group). Error bars depict SEM. **c** Tumor growth curves: control (blue), LB-100 (red), aPD-1 (green), and combination (purple). Individual and mean tumor volume over time. **d** Cumulative survival of mice over time. **e**, **f** Effect of LB-100 is dependent on adaptive immunity. Immunodeficient NSG mice without T, B, and NK cells were similarly inoculated with CT26 cells and randomized into four treatment groups as in **a**. **e** Tumor growth curves with individual and mean tumor volume over time. **f** Cumulative survival of mice over time. **g–i** Synergy of LB-100 with PD-1 blockade is dependent on CD8+ T cells. BALB/c mice were inoculated similarly as above. **g** When tumors reached 30–100 mm³, mice were temporarily depleted of CD8+ T cells and treated with combination. **h** Tumor growth curves: control (blue), combination (purple), CD8 depletion only (green), and combination with CD8 depletion (red). Individual and mean tumor volume over time. **i** Cumulative survival of mice over time. Data are representative of two independent experiments. *$P < 0.05$, **$P < 0.01$, and ****$P < 0.0001$ (log-rank test)

was increased 13-fold 10 days after treatment in the combination group with CD8 depletion compared to tumor volume in the non-depleted group (612 vs 46 mm$^3$, $p < 0.001$ by one-way ANOVA, Tukey's multiple comparison test). Survival was also significantly decreased with CD8 depletion ($p < 0.0001$ by log-rank test). CD8 T-cell depletion alone had a small deleterious effect compared to control in both tumor growth and survival, suggesting a baseline level of CD8+ T-cell-mediated immunity served to limit CT26 growth in baseline conditions. These data indicated that LB-100 with aPD-1 synergy was dependent upon CD8+ T-cell-mediated adaptive immunity and not a direct effect of PP2A inhibition on tumor growth.

**Combination therapy induces antigen-specific long-term memory.** The hallmark of a successful adaptive immune response is the establishment of immunologic memory. We tested mice that experienced a complete response (CR) for their secondary protective antitumor immunity. Mice were re-challenged with both CT26 cells in the flank and 4T1 cells, an unrelated murine breast cancer cell line, in the mammillary fat pad (Fig. 2a). Mice with CR were resistant to CT26 but not to 4T1 cells (Fig. 2b).

Eighteen days after inoculation, there was no difference in 4T1 tumor volume between naive and CR mice, while CT26 failed to grow in CR mice (Fig. 2c). This result indicated that combination-treated mice were able to develop secondary memory response that was antigen-specific to CT26 tumors.

**Enhanced activation of lymphocytes with combination therapy.** To address the cellular mechanism mediating tumor rejection by the LB-100/aPD-1 combination, we examined the status of the immune system in the secondary lymphoid organs and in the tumor. Mice were implanted with CT26 tumors and treated with LB-100 and/or aPD-1 as described above. On day 3, after two treatments, the spleens, tumor draining lymph nodes (dLN) and tumors were harvested and analyzed by flow cytometry (Figs. 3 and 4). In the secondary lymphoid tissue, we observed a greater activation of CD8+ T cells in mice treated with the combination regimen compared to controls, as indicated by greater frequency of CD44+CD62L−CD8+ T cells (Fig. 3a–c). In the spleen, treatment with LB-100 alone resulted in a small increase in CD44+CD62L−CD8+ T cells (from 13.0 to 16.6%, $p < 0.05$ by one-way ANOVA, Tukey's multiple comparison test) but the

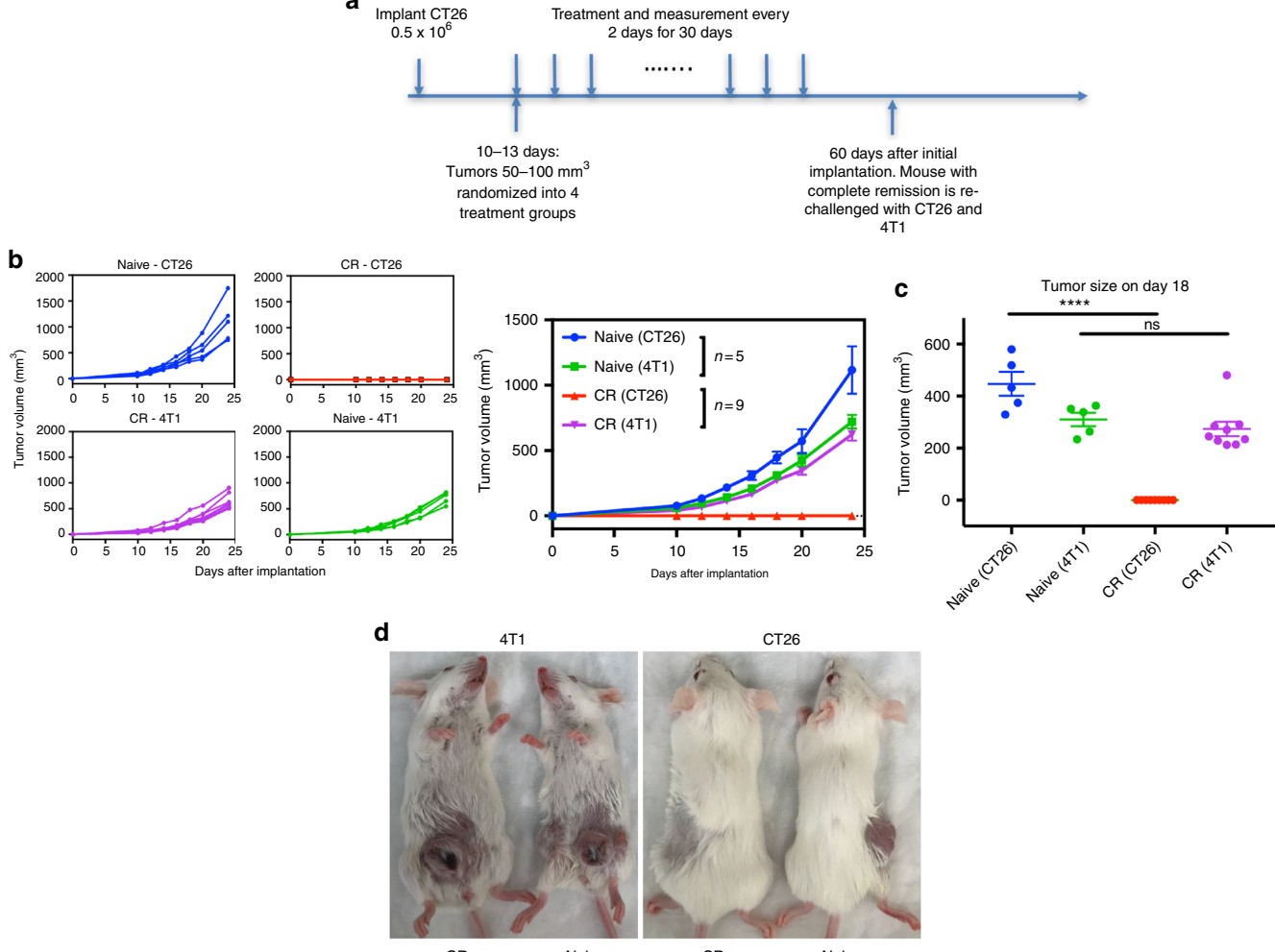

**Fig. 2** PP2A inhibition and PD-1 blockade result in a long-term antitumor antigenic-specific memory of cured animals. **a** BALB/c mice were inoculated with $0.5 \times 10^6$ CT26 cells subcutaneously and treated as above. CR or naive control mice were re-challenged about 60-days after initial implantation with $0.5 \times 10^6$ CT26 cells in the thoracic flank in combination with $1.25 \times 10^5$ 4T1 breast carcinoma cells in the mammillary fat pad. **b, c** Naive—CT26 (blue), CR—CT26 (red), naive—4T1 (green), CR—4T1 (purple). **b** Individual and mean tumor growth curves over time. **c** Quantitation of CT26 and 4T1 tumor volume 18 days after inoculation. ($P < 0.0001$, one-way ANOVA with Tukey's multiple comparison test). **d** Picture of representative naive and CR mouse following inoculation of CT26 and 4T1 tumors

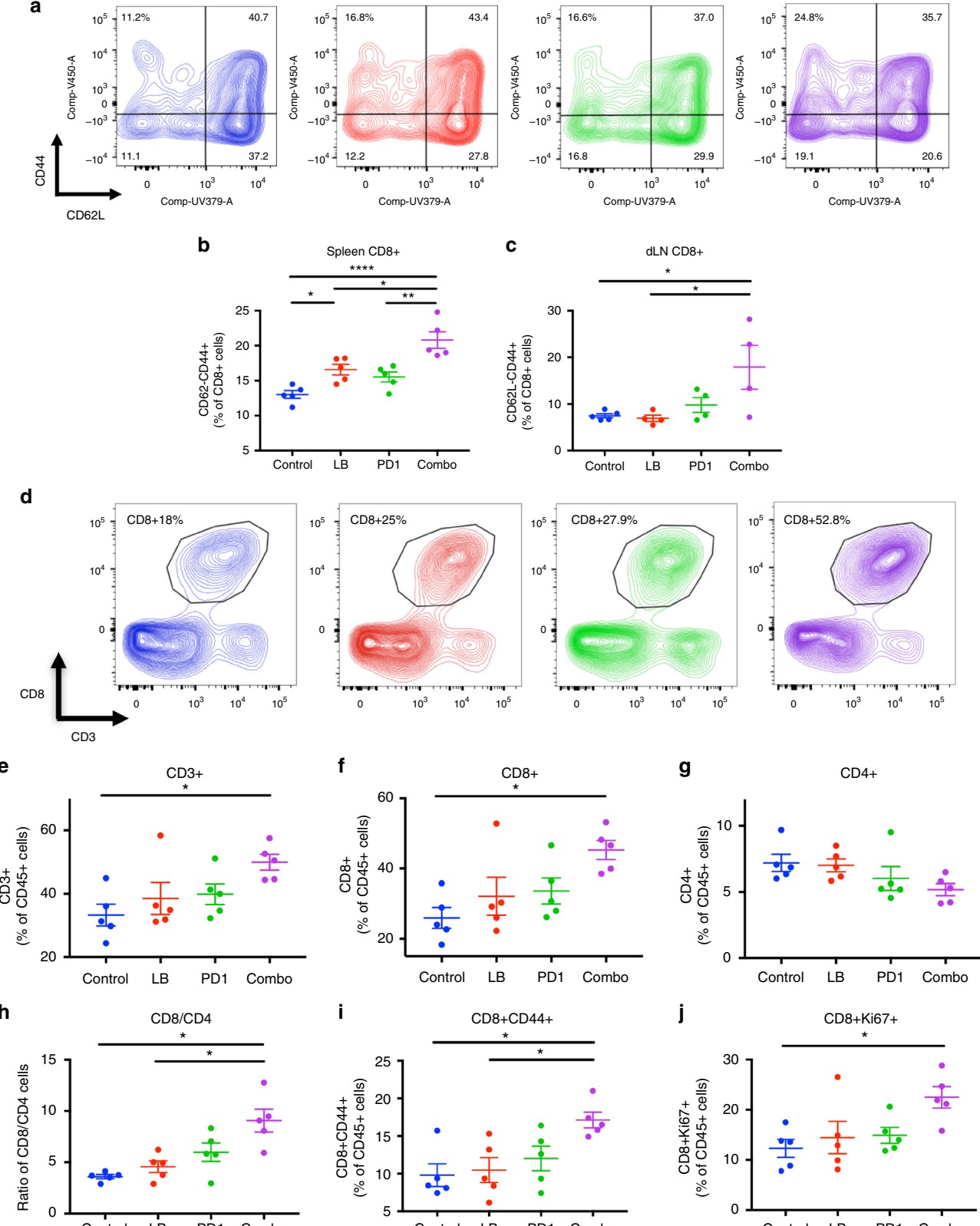

Fig. 3 PP2A inhibition and PD-1 blockade increase CD8+ activation in lymphoid organs and CD8+ infiltration in tumors. Balb/c mice in the respective groups were harvested 3 days after initiation of treatment and the spleen, tumor draining lymph nodes (dLN) and tumors were analyzed by flow cytometry. **a** Representative FACS plots of CD44 and CD62L in CD8+ T cells in the spleen. **b** Quantification of CD62-CD44+ (of CD8+ T cells) in the spleen and **c** tumor draining lymph nodes ($n = 4$–5). **d** Representative FACS plots of CD8+ CD3+ T cells as percentage of CD45+ cells. **e–g** Immune infiltrate analysis of **e** CD3+, **f** CD8+, and **g** CD4+ expressed as percentage of CD45+ cells ($n = 5$). **h** Ratio of CD8+ to CD4+ cells in tumor. **i** CD8+ CD44+ and **j** CD8+ Ki67+ expressed as percentage of CD45+ cells in tumor. *$P < 0.05$, (one-way ANOVA with Tukey's multiple comparison test). Error bars depict SEM. Data represent one of two experiments with five independently analyzed mice/group

combination treatment resulted in a greater increase than either LB-100 or aPD-1 alone (20.8 compared to 16.6 and 15.5% respectively, $p < 0.05$ and $p < 0.005$ by one-way ANOVA, Tukey's multiple comparison test) (Fig. 3b). Similarly, CD44+CD62L−CD8+ T cells were increased in the dLN of mice treated with the combination compared to control (from 7.4 to 17.9%, $p < 0.05$ by

one-way ANOVA, Tukey's multiple comparison test) (Fig. 3c). There was no difference in frequency of CD44+CD62L− subset in CD4+ T cells in both the spleen and dLN (Supplementary Fig. 1a and Supplementary Fig. 2a). Immune checkpoint markers, including expression of CTLA-4, TIM3, and Ox40 on CD8+ and CD4+ T cells were examined in the tumor draining lymph node

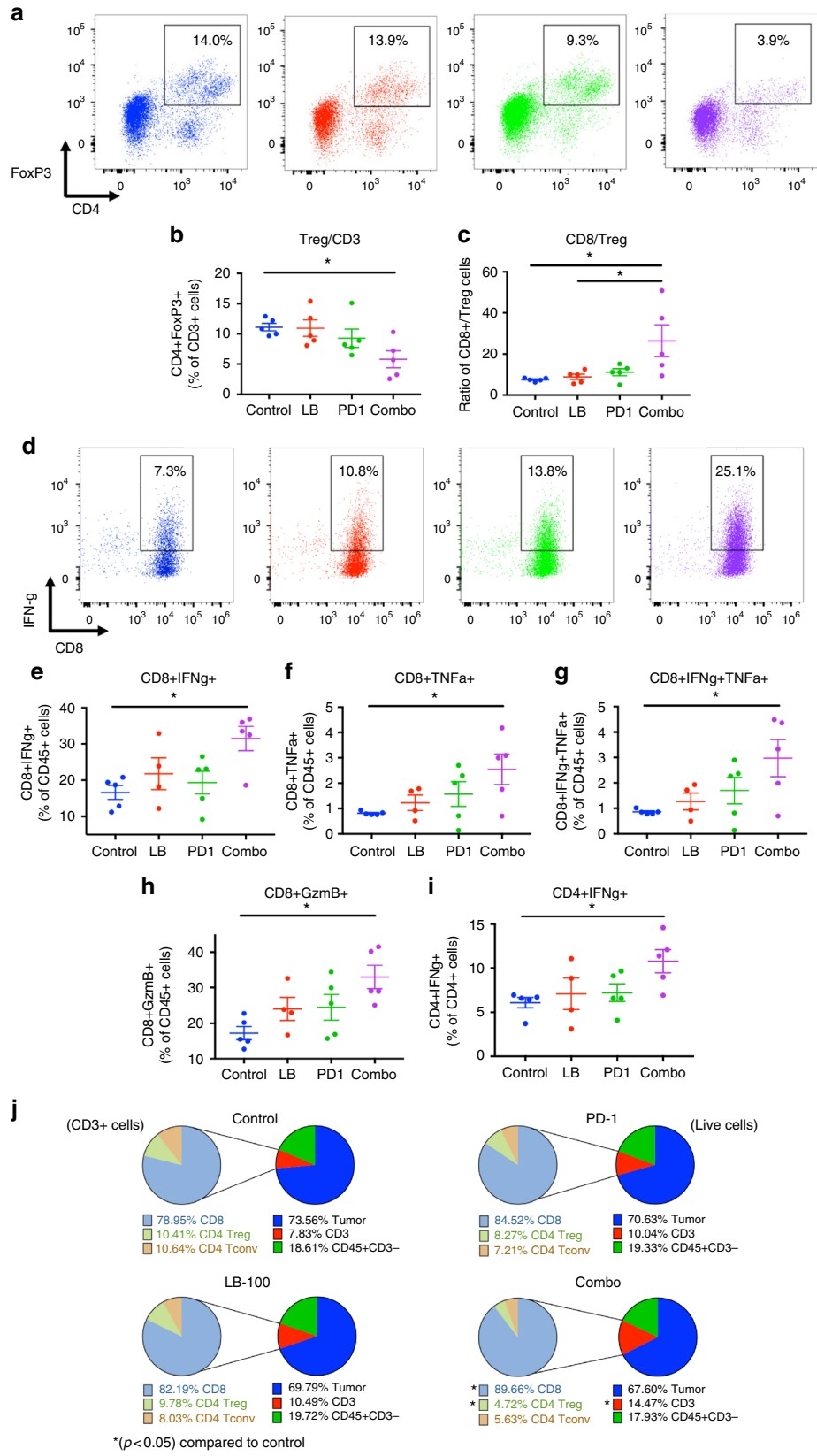

and spleen (Supplementary Fig. 1 and Supplementary Fig. 2). There was no difference in the expression of these markers.

We then performed a comprehensive analysis of the tumor infiltrating lymphocytes (Figs. 3 and 4, Supplementary Fig. 3). First, we examined the absolute percentage of CD45+ cells. There was no significant difference among the treatment groups. However, within the CD45+ population, there was a significant increase in CD3+ T cells in the combination treated group compared to control (from 33.3 to 49.9%, $p < 0.05$ by one-way ANOVA, Tukey's multiple comparison test) (Fig. 3e). More importantly, this increase in CD3+ T-cell population was attributed to a significant increase of CD8+ T cells (Fig. 3d), whether normalized to CD45+ cells (from 25.9 to 45.3%, $p \leq 0.01$ by one-way ANOVA, Tukey's multiple comparison test) (Fig. 3f) or number of tumor-resident cells (from 8 to 19%, $p < 0.05$ by one-way ANOVA, Tukey's multiple comparison test) (Supplementary Fig. 4a). A similar trend was observed in CD8+ T cells normalized to tumor weight (Supplementary Fig. 4b). Instead, the CD4+ T-cell population remained unchanged (Fig. 3g, Supplementary Fig. 4), resulting in a marked increase in CD8/CD4 ratio (from 3.6 to 9.0, $p < 0.001$ by one-way ANOVA, Tukey's multiple comparison test) (Fig. 4h). This indicated that LB-100/aPD-1 combination resulted in enhanced CD8+ T cells trafficking to the tumor, which has consistently been shown to be one the most important predictors of response to immunotherapy[18]. We further examined the subpopulation of CD8+TILs by labeling the effector phenotype marker CD44+. There was a significant increase in CD8+CD44+T cells in the combination group relative to control (9.8 to 17.1%, $p \leq 0.01$ by one-way ANOVA, Tukey's multiple comparison test) (Fig. 3i). We also found increased proliferation of CD8+TILs, as measured by expression of cell cycle associated protein Ki67 (from 12.3 to 22.5%, $p < 0.05$ by one-way ANOVA, Tukey's multiple comparison test) (Fig. 3j). Next, we examined the expression of an array of immune checkpoint markers in the TILs, including TIM3, Ox40, CTLA-4, and LAG-3. Expression of these markers were not changed with single or combination treatment (Supplementary Fig. 5), suggesting that there is potential in combining LB-100 with targeted therapeutics against these checkpoint markers.

Given the previous study demonstrating that PP2A serves an essential role in suppressive Treg[8], we examined whether addition of LB-100 could result in relative reduction of Treg cells, similar to the effect of anti-CTLA-4 therapy. aPD-1 is known to act at the level of the tumor with limited ability to deplete Tregs[3]. However, with addition of LB-100, the combination treatment significantly decreased the percentage of CD4+FoxP3+ Treg cells among TILs (from 10.3 to 4.9% of CD3+ T cells, $p < 0.05$) (Fig. 4a, b). The concomitant decrease in Treg and increase in CD8+ T cells resulted in a dramatic increase in the CD8+ to Treg ratio by 3.5-fold among the TILs (from 7.5 to 26.4, $p < 0.05$ by one-way ANOVA, Tukey's multiple comparison test) (Fig. 4c). Subsequently, we assessed the functional consequence of LB-100/aPD-1 combination in TILs. We analyzed intracellular expression of IFN-γ in response to in vitro stimulation with PMA/ionomycin. Combination treatment significantly enhanced IFN-γ production

by CD8+TILs relative to control (from 16.6 to 31.5% of CD45+, $p < 0.05$ by one-way ANOVA, Tukey's multiple comparison test) (Fig. 4d, e). In addition, the frequency of tumor necrosis factor α (TNF-α)-producing (Fig. 4f, Supplementary Fig. 6a) and IFN-γ/TNF-α dual producing (Fig. 4g, Supplementary Fig. 6b) CD8+ TILs were significantly increased with the combination treatment. The cytolytic capacity of CD8+TILs was also determined by Granzyme B (GzmB) expression, which was also significantly increased with combination treatment (Fig. 4h, Supplementary Fig. 6c). In CD4+ T cells, we observed a small, but statistically significant increase in IFN-γ production (from 6.1 to 10.8% of CD4+ cells, $p < 0.05$ by one-way ANOVA, Tukey's multiple comparison test) (Fig. 4i). This suggests that while there is no overall increase in CD4+ infiltration with LB-100/aPD-1 combination, effector CD4+ T cells present in the tumor were nonetheless more functionally active with enhanced IFN-γ production.

Taken together, combining LB-100 with aPD-1 blockade resulted in a significant change in the composition of TILs (Fig. 4j). While the overall CD45+ population remained relatively stable, there was a marked increase in CD3+ T-cell infiltration, driven by a preponderance of CD8+ T cells. At the same time, the Treg population was relatively reduced resulting in a marked increase in CD8/Treg ratio. In addition, CD8+ T cells were more proliferative and functionally active as indicated by cytokine expressions. These findings are consistent with the observation that LB-100/aPD-1 combination could elicit durable tumor rejection in CT26 in an immune-dependent manner.

### LB-100 and aPD-1 enhance antitumor activity in B16 melanoma.

We next sought to determine whether LB-100/aPD-1 combination is effective against other aPD-1 resistant tumor. In a tumor prevention model, 6–8 weeks old C57BL/6 mice were randomized into four treatment groups: PBS, LB-100, aPD-1, and combination. B16F10 cells ($2.5 \times 10^5$) were inoculated 2 days after initiation of treatment subcutaneously in the right thoracic flank. Treatments were administered every two days until mice reached survival endpoints (Fig. 5a). By day 15 after tumor implantation, there was no difference between control and the single-agent arms. However, tumor size was significantly smaller in the combination group relative to control (from 305.9 to 109.0 mm³, $p < 0.05$ by one-way ANOVA, Tukey's multiple comparison test) (Fig. 5b, c) and survival was prolonged by the combination treatment ($p < 0.05$ by log-rank test) (Fig. 5d).

It is noteworthy that none of the mice in the combination group demonstrated any clinical signs of autoimmune inflammatory events. However, given that the LB-100/aPD-1 combination resulted in increased effector function and Treg reduction, autoimmunity is a significant concern. We, therefore, examined the histology of multiple organs of treated mice to look for signs of inflammation. C57BL/6 mice that reached survival endpoints were sacrificed and the histology of the skin, salivary gland, pancreas, lung and stomach were examined (Fig. 5e, Supplementary Fig. 7). There was no evidence in any of the treatment group

**Fig. 4** PP2A inhibition and PD-1 blockade reduce Treg and enhance CD8 effector function. CT26 tumors in the respective groups were harvested 3 days after initiation of treatment and analyzed by flow cytometry. **a** Representative FACS plots of FoxP3+ and CD4+ T cells in tumors. **b** Percentage of CD4+ FoxP3+ T cells of total CD3+ cells. **c** Ratio of CD8+ to CD4+ FoxP3+ Treg cells in tumor ($n = 5$). **d–i** Harvested TILs were stimulated for 5 h with PMA/Ionomycin ex-vivo and production of intracellular cytokine was determined by flow cytometry. **d** Representative FACS plots of CD8+ IFNγ+ T cells of CD45+ cells. **e–g** Percentage of **e** CD8+ IFNγ+, **f** CD8+ TNFα+ and **g** CD8+ double positive IFNγ+ TNFα+ T cells of CD45+ cells. **h** Percentage of CD8 + Granzyme B+ T cells of CD45+ cells. **i** Percentage of CD4+ IFNγ+ of CD4+ T cells. **j** Summary of CD45+ immune cell subsets and CD45− cells as determined by FACS. Subsets are depicted as percentage of all acquired live events (right) and CD3+ cells (left). Diagram on the right: CD45− (blue), CD3 + (red), CD3− CD45+ leukocytes (green). Diagram on the left: CD8+ (light blue), CD4+ FoxP3+ Treg (light green), CD4+ FoxP3− conventional T cells (tan). *$P < 0.05$, (one-way ANOVA with Tukey's multiple comparison test). Error bars depict SEM. Data represent five independently analyzed mice/group

to suggest increased lymphocyte infiltration or signs of autoimmunity.

**LB-100 enhances antigen-specific T-cell cytotoxicity in vitro**. Previous studies suggested that inhibiting PP2A has a direct effect on CD8+ T cells by enhancing antigen-specific cytotoxic T lymphocyte (CTL) responses[4–6]. While our previous in vivo data demonstrated that LB-100 mediated antitumor activity was CD8+ T cells dependent, we have not shown evidence that CTL response was directly enhanced. To investigate if LB-100 could enhance Ag-specific CTL effector function, we performed flow

cytometry-based cell-mediated cytotoxicity[19] using OT-1 CTLs that recognized the ova peptide in B16-Ova cells. OT-1-CD8+ T cells were isolated from OT-1 TCR transgenic mice and activated with CD3/CD28 for 3 days. The activated CTLs were then pre-treated with a dose titration of LB-100 for 4 h before being mixed with B16-Ova target cells (TCs) that were pre-labeled with the florescent dye 3,3′-dioctadecyloxacarbocyanine (DiO). Following 3 h of incubation in the presence of propidium iodide (PI), cells were analyzed by flow cytometry. Dead TCs were DiO+PI+. The percentage (%) of effector cell induced cell death was calculated to correct for the rate of spontaneous TC death. Using an

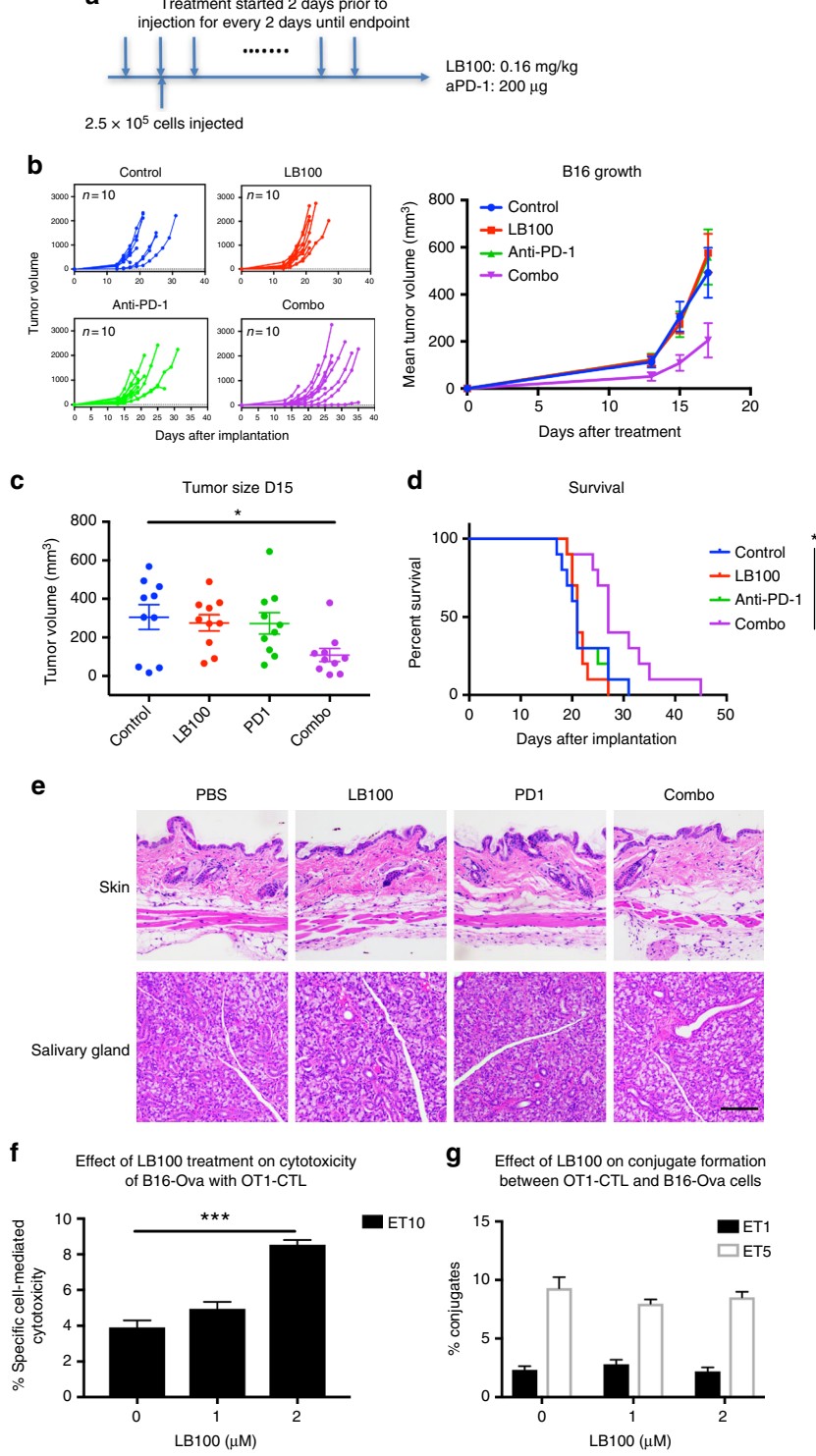

effector to target ratio (ET) of 10, the % of Ag-specific cell-mediated cytotoxicity was significantly increased when the CTLs were pre-treated with 2 µM of LB-100 compared to control (3.9% vs 8.5%, $p = 0.0002$ by one-way ANOVA, Tukey's multiple comparison test, Fig. 5f). We, thereby, demonstrated that LB-100 has a direct effect on CTLs and its Ag-specific effector function was enhanced. Next, we asked if LB-100 would change the level of CTL-TC conjugate formation. OT-1-CD8+ T cells were similarly isolated and activated as above. Cells were then pre-treated with LB-100 for 5 h and labeled with green florescent CMFDA dye. B16-Ova TCs were labeled with orange florescent CMRA dye. The CTLs and TC were then mixed and incubated for 1 h before analyzed by FACS. Conjugate formation was determined as % of CMFDA+CMRA+ double positive cells. LB-100 at concentration up to 2 µM did not affect the degree of conjugate formation (Fig. 5g) at ET of 1 or 5. This result suggests that the effect of LB-100 on CTLs is not by enhancing TCR/antigen interactions to increase effector-target cell conjugate formation but by promoting cytotoxicity of TC after conjugation formation. This is consistent with previous data showing that PP2A inhibition did not alter surface expression of CTL accessory molecules[4] and that PP2A mediated downstream signaling of CTLA-4[5].

**LB-100 inhibits PP2A activity and enhances mTORC1 activation.** To further investigate the mechanisms of LB-100 mediated antitumor activity, we performed a series of in vitro experiments. While LB-100 has previously been shown to be a selective inhibitor of PP2A[20] in multiple different tissues and cell types[21–25], none has studied its effect on lymphocytes. We have demonstrated that PP2A given at the in vivo dose of 0.16 mg kg$^{-1}$ inhibited PP2A activity in CD3+ T cells (Fig. 1b). We once again confirmed this effect in vitro in isolated CD4 and CD8 cells from mouse splenocytes measured 3 h after stimulation with plated CD3 and soluble CD28. Consistent with studies in other cells, there was a dose-dependent decrease in PP2A enzymatic activity in both CD4 and CD8 cells with a greater effect in CD8 than CD4 cells (Fig. 6a). Previous study has shown that activation of PP2A in lymphocytes results in a distinct inhibitory effect on mTORC1 activity but sparing mTORC2 and PI(3)K-AKT pathways[8]. We, therefore, hypothesized that LB-100 mediated PP2A inhibition could result in specific activation of the mTORC1 axis. After 3 h of in vitro activation of isolated CD3 cells, we assessed the activity of mTORC1, mTORC2, and PI(3)K-AKT pathways by checking the phosphorylation of ribosomal S6 protein (S6), AKT at Thr473, and AKT at Thr308, respectively. We found that while LB-100 has minimal effect on mTORC2 and PI(3)K-AKT pathways (Fig. 6b), there was a dose-dependent increase in activity of mTORC1 as measured by phosphorylation of S6 (Fig. 6c). This difference was not observed in any of the three pathways at

an early time point of 30 min after activation (Supplementary Fig. 8).

**LB-100 inhibits development of regulatory or Th2 CD4 cells.** It is well established that mTORC1 signaling plays a critical role in determining lineage commitment of T lymphocytes[26]. Hyperactivation of mTORC1 in naive CD4 T cells have been shown to prevent Treg development[27]. In addition, mTORC1 activity was found to be essential for Th1 developemnt from naive CD4+ cells[28]. Given the observation that LB-100 preferentially hyperactivated mTORC1 signaling, we postulated that LB-100 could inhibit Treg and Th2 CD4+ cells development in their respective skewing conditions.

Naive CD4+ cells were isolated from mouse splenocytes and activated in vitro with anti-CD3 and CD28 in the presence of TGF-β or IL4 to induce development of Treg or Th2 CD4+ cells, respectively. After 72 h, intranuclear expression of Foxp3 or GATA3 was quantitated by flow cytometry to determine percentage of Treg or Th2 cells respectively. LB-100 treatment significantly impaired induction of Foxp3 by TGF-β (Fig. 7a) or GATA3 (Fig. 7b) by IL4 in a dose-dependent manner. In addition, we quantified the relative proportion of Th2 and Th1 CD4+ cells. The frequency of GATA3 expressing cells relative to T-bet expressing cells decreased significantly with LB-100 treatment under Th2 skewing condition (Fig. 7c). We then explored the functional consequence of Th1 CD4+ cell with LB-100 treatment. Under both Th1 and Th2 skewing conditions, there was a dose-dependent increase in IFN-γ expression with PP2A inhibition. This was shown with both intracellular staining (Fig. 7d) and measurement of cytokine secretion (Fig. 7e, f). Other Th1-related cytokines, including TNF-α and IL2 were also increased in both Th1 and Th2 conditions (Fig. 7e, f). Secretion of IL4 was decreased under Th2 condition (Fig. 7f). These data suggest that PP2A inhibition decreased Treg formation and skewed CD4 cells differentiation towards Th1 lineage resulting in an overall increase in Th1 cytokine secretion. The results of these in vitro experiments are consistent with the in vivo TILs findings and potentially suggests that PP2A inhibition enhanced cancer immunity via mTORC1 hyperactivation.

**LB-100 in human mixed lymphocytes reactions.** To further confirm that the immune-modulating effect of LB-100 has clinical utility, we performed mixed lymphocyte reactions (MLRs) using PBMC from healthy human donors. Monocyte-derived dendritic cells were co-cultured with allogenic CD4+ T cells labeled with cytosolic dye CFSE. LB-100 was given on the day of co-culture (day 0) and again on day 3. Proliferation and IFN-γ secretion by CD4 T cells were assessed on day 5 (Fig. 8a). There was a significant increase in CD4 T-cell proliferation, as measured

**Fig. 5** PP2A inhibition and PD-1 blockade synergistically decrease tumor growth of B16 melanoma. **a** C57BL/6 mice were randomized into four treatment groups. $2.5 \times 10^5$ B16F10 cells were inoculated 2 days after initiation of treatment subcutaneously in the right thoracic flank. Mice were treated every two days until survival end point. **b** Tumor growth curves: control (blue), LB-100 (red), aPD-1 (green), and combination (purple). Individual and mean tumor volume over time. **c** Quantitation of B16 tumor volume 15 days after inoculation. (*$P < 0.05$, one-way ANOVA with Tukey's multiple comparison test). **d** Cumulative survival over time. *$P < 0.05$, (log-rank test). **e** At survival end point, organs of the mice were harvested for fixation and staining. Representative images of hematoxylin and eosin staining of the skin and salivary gland of each treatment group ($n = 2$ per group). Scale bars, 100 µm. **f** LB-100 enhanced Ag-specific CTL cytotoxicity. OT-1 CD8 cells were isolated from spleens of OT-1 TCR transgenic mice and activated with CD3/CD28 for 3 days. CTLs were treated with LB-100 for 4 h before being mixed with DiO-labeled B16-Ova target cells (TCs). After 3 h of incubation in the presence of PI, cells were analyzed by flow cytometry. Dead TCs were DiO+ PI+. % of effector cell induced cell death was calculated. Ag-specific cell-mediated cytotoxicity was significantly increased when the CTLs were pre-treated with 2 µM LB-100 compared to control. (**$P < 0.01$, one-way ANOVA with Tukey's multiple comparison test). **g** LB-100 did not change level of conjugation formation. OT-1 CD8 cells were similarly isolated and activated as above. Cells were then pre-treated with LB-100 for 5 h and labeled with green florescent CMFDA dye. B16-Ova TCs were labeled with orange florescent CMRA dye. The CTLs and TC were then mixed and incubated for 1 h before analyzed by FACS. Conjugate formation was determined as % of CMFDA+ CMRA+ double positive cells. LB-100 at concentration up to 2 µM did not affect the degree of conjugate formation using both ET of 1 and 5. Data are from one experiment representative of two independent experiments with similar results

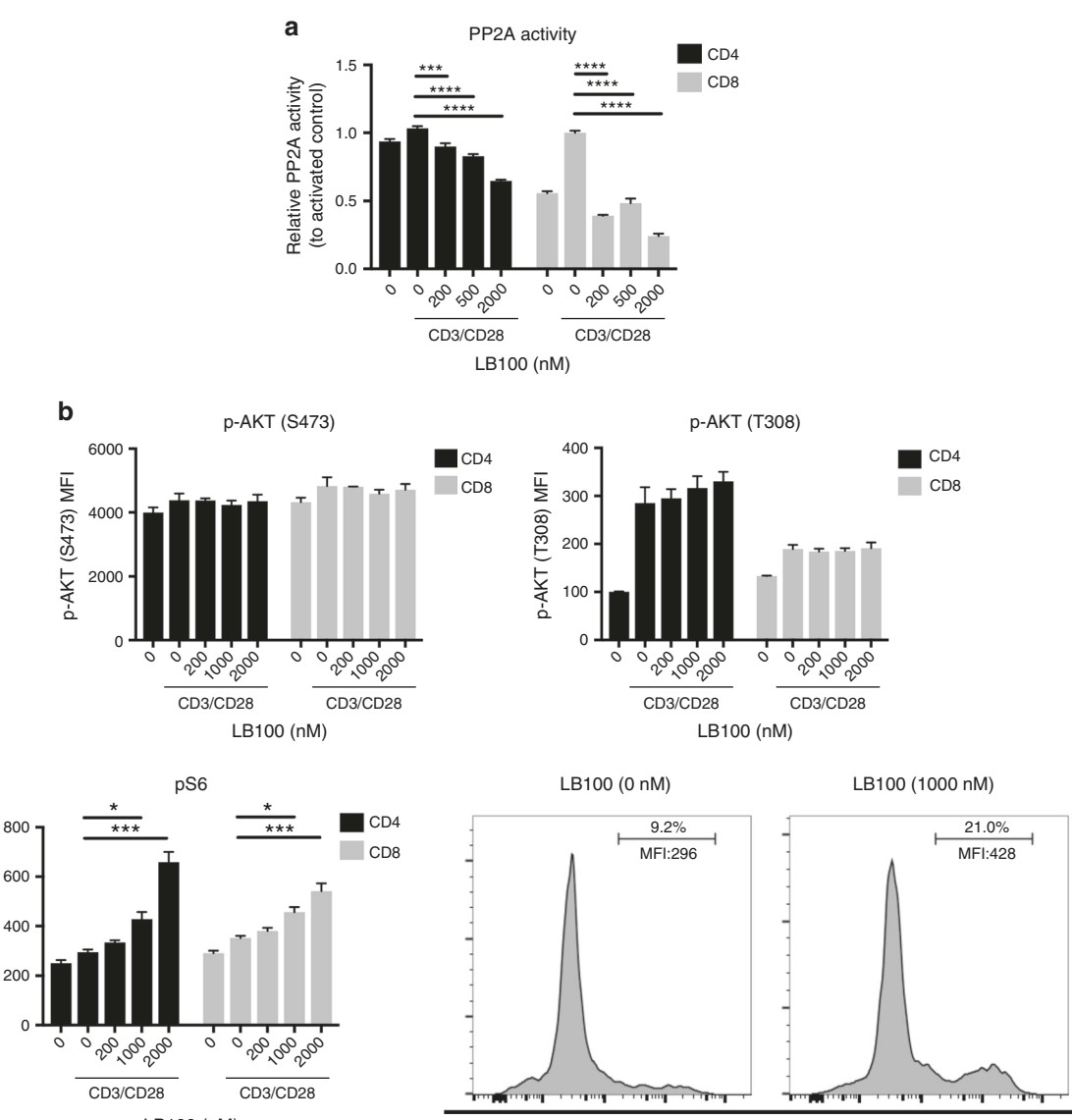

**Fig. 6** LB-100 inhibits PP2A enzymatic activity and activates the mTORC1 pathway. CD3 T cells were isolated from mice splenocytes and cultured with or without stimulation using immobilized anti-CD3 (10 μg ml$^{-1}$) and soluble anti-CD28 (2 μg ml$^{-1}$). **a** PP2A enzymatic activity was measured after 3 h of activation. PP2A activity was measured as relative to activated control in presence of LB-100 dose titration. **b** Flow cytometry analyzing AKT phosphorylated at Thr308 (p-AKT(T308)) or Ser473 (p-AKT(S473)) after 3 h of stimulation in presence of LB-100 dose titration. **c** Flow cytometry analyzing phosphorylated S6 (p-S6) in presence of LB-100 dose titration. *$P < 0.05$, ***$P < 0.001$ (one-way ANOVA with Tukey's multiple comparison test). Data are from one experiment representative of two independent experiments with similar results. Error bars depict SEM

by the percentage of dividing cells, with LB-100 treatment at 1 μM (Fig. 8b). There was also a trend towards increased proliferation at lower LB-100 concentrations (in the sub-micromolar range). At the high dose of 5 μM, proliferation was impaired suggesting that there is an optimal window of LB-100 exposure that enhanced immunity. A similar pattern was observed with IFN-γ secretion (Fig. 8c). At 0.2 and 1 μM of LB-100, IFN-γ release was significantly enhanced 3.5- to 4-fold, respectively. We also examined the effect of lineage differentiation in CD4+ T cells by labeling for T-bet. LB-100 at 1 μM significantly increased T-bet expression (Fig. 8d), confirming our previous finding that LB-100 skewed CD4 lineage towards Th1 differentiation We then tested whether LB-100 could enhance IFN-γ secretion in vitro in combination with aPD-1 blockade using Nivolumab. A similar MLR assay was performed, and we found that LB-100 synergized with aPD-1 blockade and enhanced IFN-γ secretion compared to single agents (Fig. 8e).

## Discussion

The advent of checkpoint inhibition with monoclonal antibodies against PD-1/PD-L1 and CTLA-4 represents a milestone for anticancer immunotherapy. Clinical trials demonstrated dramatic responses in various types of advanced cancer[29], for which alternative treatments are lacking. However, it is also clear that mono-immunotherapy is only efficacious in a small subset of patients, even in tumors that are inherently immunogenic[30]. The combination of aPD-1 with aCTLA-4 blockade resulted in significant increase in survival but is associated with high prevalence of autoimmunity-related adverse events[31]. Rational combinations to target non-redundant pathways of aPD-1/PDL-1 resistance while minimizing autoimmunity is critical to fully realize the potential of checkpoint immunotherapy. Combining immune checkpoint blockade with small molecule therapies is an area of great translational interest. Pre-clinical studies have shown synergistic potential by combining aPD-1 blockade with

Indoleamine 2,3-dioxygenase (IDO) inhibitors[32], Bruton's tyrosine kinase inhibitors[16], and MAP kinase inhibitors[33] among others. To our knowledge, this is the first report demonstrating that pharmacologic inhibition of PP2A results in enhancement of immune-mediated antitumor activity when combined with aPD-1

blockade. In CT26 tumor-bearing mice, co-inhibition of PP2A and PD-1 was required to elicit tumor rejection and antigen-specific protection against tumor re-challenge. This effect is CD8+ T-cell dependent and TILs analysis demonstrated markedly increased infiltration of CD8+ T cells with enhanced effector

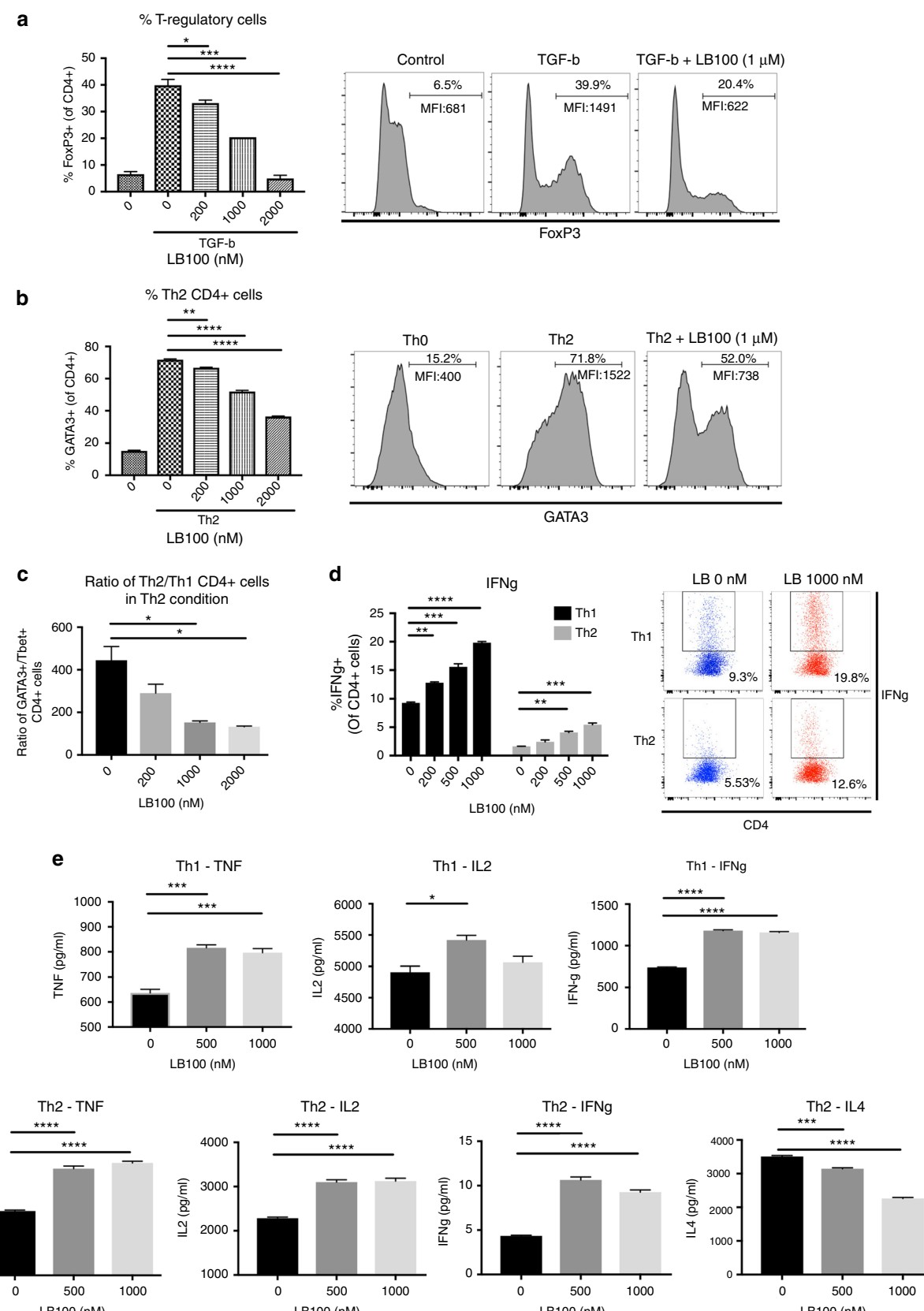

**Fig. 7** LB-100 impairs induction of naive CD4+ T cells into Treg or Th2 lineage. Naive CD4+ T cells were isolated from mice splenocytes and stimulated with anti-CD3 (10 μg ml$^{-1}$) and soluble anti-CD28 (2 μg ml$^{-1}$) under different skewing conditions. After 3 days in culture, intracellular protein expression was analyzed by flow cytometry and cytokines secretion was measured by bead-based multianalyte flow immunoassays. LB-100 was replenished daily with the indicated concentration. **a** Naive CD4+ cells were induced into Tregs in the presence of TGF-β with titration of LB-100 for 3 days. Intracellular Foxp3 was labeled with flow cytometry. Quantification of CD4+ Foxp3+Treg cells and representative flow cytometry data demonstrated a decrease in % of Foxp3+ CD4+ cells with LB-100. Cells were gated on CD4+ cells. **b, c** Naive CD4+ cells were induced into Th2 CD4+ cells in the presence of IL4 with titration of LB-100 for 3 days. **b** Intracellular GATA3 was labeled with flow cytometry. Quantification of CD4+ GATA3+ Th2 cells and representative flow cytometry data demonstrated a decrease in % of GATA3+ CD4+ cells with LB-100. Cells were gated on CD4+ cells. **c** Ratio of GATA3+ Th2 over T-bet+ Th1 CD4+ cells under Th2 condition was decreased with LB-100. **d** Purified naive CD4+ T cells were induced in the presence of Th1 or Th2 skewing condition. On day 3, cells were stimulated and intracellular production of IFN-γ was measured by flow cytometry. Quantification of CD4+ IFN-γ+ cells and representative flow cytometry data demonstrated an increase in % of IFN-γ+ CD4+ cells with LB-100 in both Th1 and Th2 conditions. Cells were gated on CD4+ cells. **e** TNF, IL2, and IFN-γ production in supernatant of naive CD4+ T cells activated in Th1 skewing conditions for 3 days. **f** TNF, IL2, IFN-γ, and IL4 production in supernatant of naive CD4+ T cells activated in Th2 skewing conditions for 3 days. Cytokine levels were adjusted to absolute cell number. *P < 0.05, **P < 0.01, ***P < 0.001 (one-way ANOVA with Tukey's multiple comparison test). Data are from one experiment representative of two independent experiments with similar results. Error bars depict SEM

and cytolytic functions. In addition, immune-suppressive CD4+ Treg cells were concomitantly reduced resulting in a markedly enhanced CD8 to Treg ratio.

The rationale for combining PP2A inhibition with immunotherapy is supported by multiple independent studies showing that inhibiting PP2A enhances T-cell activity via multiple mechanisms. Taffs et al[4]. was the first to identify PP2A as a potential negative regulator of T-cell activation through regulation of TCR mediated transmembrane signaling. Using okadaic acid, they demonstrated that PP2A inhibition could enhance cell–cell contact dependent and Ag-specific effector functions of lymphocytes. This result was later corroborated by Parry et al[5]., in which they identified PP2A as the phosphatase responsible for CTLA-4 mediated deactivation of Akt signaling. More recently, Zhou et al[6] identified PP2A as the most promising immunotherapy target in an in vivo shRNA screen. They demonstrated using a B16-ova melanoma model that directly inhibiting PP2A in adoptively transferred effector CD4+ or CD8+ cells could enhance antitumor immunity. This suggests that PP2A inhibition would have a direct effect on effector cells by enhancing their activation and proliferation. We have shown in our data that LB-100 indeed could enhance Ag-specific CTLs. To futher strengthen the case for PP2A as an immunotherapy target, Apostolidis et al[8]., showed that PP2A activity was preferentially increased in Treg due to FoxP3 downstream signaling. PP2A function was found to be essential for the immunosuppressive phenotype of Tregs. Transgenic knockdown of PP2A in Tregs resulted in increased T effector function and impaired Treg suppressive capacity. They also showed that in lymphocytes, PP2A inhibited mTORC1 signaling and transgenic knowdown of PP2A-induced specific activation of mTORC1 complex. Consistent with those findings, we demonstrated that pharmacologic inhibition of PP2A with LB-100 specifically enhanced mTORC1 activity while sparing the mTORC2 and PI(3)K-AKT pathways. The importance of mTORC1 signaling in determining lineage differentiation of CD4 T cells is well established. Hyperactivation of mTORC1 has been shown to inhibit differentiation of naive CD4+ cells into Treg and mTORC1 has been shown to be essential for Th1 differentiaion. We showed that in vitro LB-100-mediated PP2A inhibition impaired naive CD4+ cells induction into Treg and Th2 CD4+ cells in their respective skewing conditions. Additionally, there was an overall tilting of CD4+ cells towards Th1 phenotype with enhanced Th1-related cytokine production. Taken together, the data suggest that inhibiting PP2A could enhace the antitumor response of the adaptive immune system via multiple and complementary mechanisms.

PP2A is a serine/threonine phosphatase consisting of a catalytic subunit (C), a structural subunit (A) and a regulatory subunit (B)[34]. The catalytic and structural subunits have 2 families,

while the regulatory subunits have four families. Each family contains several isoforms allowing up to 96 possible combinations for the PP2A holoenzyme[35]. The expression levels of each subunit is highly variable and dependent on cell and tissue type[36] conferring the diversity of PP2A subcellular location and function. LB-100 is a small molecule analog of cantharidin. Its chemical structure has been previously described[20]. LB-100's active metabolite, endothall, has been shown to bind the catalytic subunit of the PP2A enzyme[37]. Studies using PP2A as a pharmacologic target in cancer therapy gave contradictory results[35], in which PP2A inhibition using natural compounds, such as okadaic acid and microcystins, have been shown to be both pro-proliferative and pro-apoptotic in cancer cells. It is now known that PP2A inhibition has a dose-dependent dualistic response between inducing apoptosis and cell proliferation[35,38]. In a specific context, low dose PP2A inhibition could promote cell proliferation by activating kinases such as MAPK[38,39], while high dose PP2A inhibition could induce cell death via blockage of cell cycle at G1/S[38,40,41]. The specific downstream mechanisms of PP2A inhibition in lymphocytes are less well elucidated. But we similarly observed, in our in vitro mixed lymphocyte reaction studies, a dose-dependent response in which low dose PP2A inhibition increased T-cell proliferation while high dose (>5 μM) had the opposite effect. The dosage used in our in vivo study was significantly lower (10-fold) than previously used in other pre-clinical studies administering LB-100 for chemo- and radio-sensitization[13–15]. The empirical dosage was chosen based on a series of pilot dose titration experiments in mice and the experience with one patient who had an objective tumor regression in a phase I trial with LB-100 at a low dose[12]. The human dose-equivalent used in our in vivo study is 0.5 mg m$^{-2}$, significantly below the maximally tolerated dose of 2.33 mg m$^{-2}$, suggesting that the dosage needed to elicit an immune response should be well tolerated in humans. The critical element moving forward would be to identify an optimal LB-100 dose to combine with approved aPD-1 blockade to be used in clinical trials.

Our study demonstrates that PP2A can be targeted pharmacologically to enhance immune-mediated antitumor response when combined with checkpoint inhibition, using a small molecule compound that has been shown to be safe in early clinical testing. Importantly, there was no clinical or histologic evidence of increased autoimmunity. Therefore, our results carry great translational potential and argue for clinical evaluation of LB-100 in combination with PD-1 blockade in treatment of solid tumors.

## Methods
**Drugs**. Nivolumab was obtained from Bristol-Myers Squibb and LB-100 was obtained from Lixte Biotechnology Holdings, Inc.

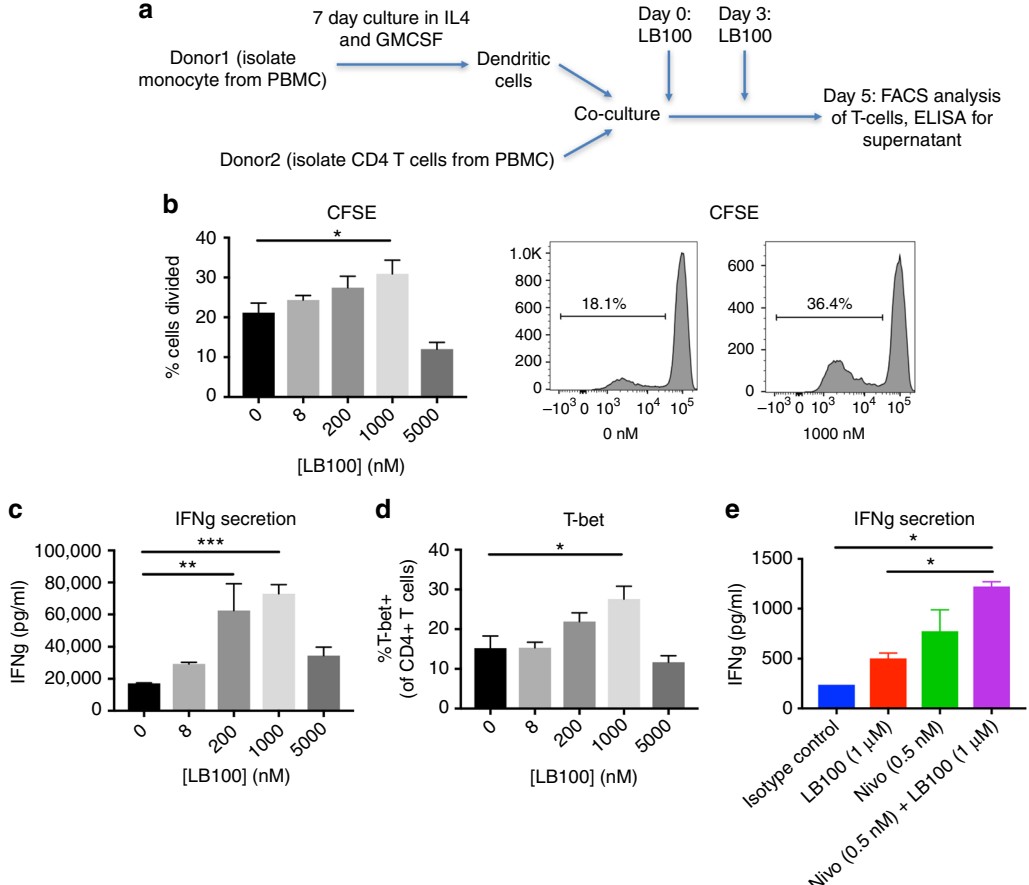

**Fig. 8** PP2A inhibition enhances T-cell function in human mixed lymphocyte reactions. **a** DCs were induced from purified monocytes by culturing in IL4 and GM-CSF for 7 days. $10^5$ purified CFSE labeled CD4+ T cells were then co-cultured with $10^4$ allogenic DCs in the presence of LB-100 titration in duplicates or triplicates for 5 days. LB-100 was replenished on day 3. Supernatant was collected on day 5 and measured for IFN-γ production. FACS analysis was performed on the cultured cells. **b** In vitro proliferation of CD4+ T cells in the presence of LB-100 dose titration, measured by dilution of the cytosolic CFSE. Percentage of cells divided was plotted against concentration of LB-100 and representative flow cytometry data demonstrated increase in % cells divided at 1 μM of LB-100. **c** IFN-γ production measured at day 5 demonstrated a dose-dependent increase in IFN-γ secretion with LB-100. **d** Intracellular staining of T-bet was performed in CD4+ T cells after 5 days of co-culture. Percentage of CD4+ T-bet+ (of CD4+ cells) was increased at 1 μM of LB-100 . **e** IFN-γ production in cells treated with isotype control, LB-100 and/or Nivolumab demonstrated a synergistic increase in IFN-γ production with combination treatment. 1 μM of LB-100 and 0.5 nM (75 ng ml$^{-1}$) of Nivolumab, $*P < 0.05$, $**P < 0.01$, $***P < 0.001$ (one-way ANOVA with Tukey's multiple comparison test). Data are from one experiment representative of two independent experiments with similar results. Error bars depict SEM

**Cell lines**. CT26.CL25 colon carcinoma, B16F10 melanoma and 4T1 mammary carcinoma cell lines were obtained from ATCC. B16-Ova melanoma cells were provided by Dr. Gattinoni (National Institutes of Health, Bethesda, MD). Tumor cells were cultured in complete medium (RPMI 1640, Gibco) containing 10% (vol/vol) FBS (Gibco), 100 U ml$^{-1}$ penicillin, 100 μg ml$^{-1}$ streptomycin (Gibco). We verified that none of the cell lines used in this study were found in the Register of Misidentified Cell Lines maintained by the International Cell Line Authentication Committee (http://iclac.org/databases/cross-contaminations/). All cell lines used were tested and shown to be negative for mycoplasma contamination using PCR amplification

**Syngeneic tumor models**. Mice were maintained and experiments were conducted with the approval of the NINDS and NCI Animal Use and Care Committees. For CT26 tumors: Female BALB/c (6–8 week old) were purchased from Charles River Laboratory. CT26 cells ($0.5 \times 10^6$) were injected into the right flank subcutaneously. Once tumors reached a volume of 50–100 mm$^3$ (day 0), mice were randomized and treated with PBS ($n = 6$), LB-100 (0.16 mg kg$^{-1}$) ($n = 6$), anti-mouse PD-1 (10 mg kg$^{-1}$) (RMP1-14; rat IgG2b; BioXcell) ($n = 5$) and combination ($n = 6$). Treatments were given every 2 days for 30 days. Tumor volume was measured every 2 days using a caliper and tumor volume was calculated according to the formula: Volume (mm$^3$) = L × W$^2$ /2, where $L$ is the length and $W$ is the width of the tumor (in millimeters). For the experiment using immuocompromised mice, male NSG mice (6–8 week old) were obtained from NCI-Frederick animal facility. CT26 tumors were similarly implanted as above. Mice were randomized when tumor reached a volume of 50–100 mm$^3$ (day 0) into four groups: PBS ($n = 8$), LB-100 (0.16 mg kg$^{-1}$) ($n = 8$), anti-mouse PD-1 (10 mg kg$^{-1}$) ($n = 8$) and

combination ($n = 8$). For the CD8 depletion study, female BALB/c (6–8 week old) mice were used. Once tumors reached 30–100 mm$^3$, mice were then randomized into 4 groups: PBS ($n = 8$), CD8 depletion ($n = 7$), combination ($n = 11$), CD8 depletion plus combination ($n = 8$). Mice in CD8 depletion groups were injected with 250 μg of CD8-depleting antibodies (clone 53.6.7; BioXcell). Injections were given 2 day, 1 day, and on the day of therapy initiation (day −2, −1, 0) then again on day 5, 8, and weekly onwards. A smaller starting tumor volume was used as criteria for randomization than previous experiments because randomization took place 2 days prior to treatment. For B16 tumors: Female C57BL/6 mice (6–8 week old) were purchased from Charles River Laboratory. Mice were first randomized into respective four treatment groups (day 0): PBS ($n = 10$), LB-100 (0.16 mg kg$^{-1}$) ($n = 10$), anti-mouse PD-1 (10 mg kg$^{-1}$) ($n = 10$) and combination ($n = 10$). On day 2, B16F10 cells ($2.5 \times 10^5$) were injected into the right flank subcutaneously. Treatment and measurements were done every 2 days. Survival end point for all animal studies were defined as when any of the following criteria was reached: (1) tumor volume exceeding 2000 mm$^3$, (2) tumor diameter exceeding 2 cm, (3) severe non-healing skin necrosis over the tumor.

Sample sizes of animal studies were determined empirically based on previous pilot experiments. Pre-established starting tumor size used as inclusion criteria for randomization was determined by pilot experiments. Investigator performed randomization manually and was not blinded from the group allocation during outcome measurement.

For tumor re-challenge study, mice with CR to combination treatment ($n = 9$) and age-matched naive female BALB/c (16–20 week old) ($n = 5$) were inoculated with $0.5 \times 10^6$ CT26 cells into flank and with $1.25 \times 10^5$ 4T1 mammillary carcinoma cells in the contralateral mammary fat pad. Tumor volumes were then monitored similarly as above.

**Isolation of TILs**. Mice were injected in the right thoracic flank with $0.5 \times 10^6$ CT26 cells and treated as above after tumors reached between 50 and 100 mm³. After 2 treatments, mice were sacrificed and tumors excised. Tumors were subjected to mechanical disruption using a GentleMACS Dissociator (Miltenyi Biotec) in presence of enzymatic digestion using Tumor Dissociation Kit (Miltenyi Biotec). Gating strategy used for analysis of TILs is shown in Supplementary Fig. 3.

Intracellular cytokine staining, phosphoflow and flow cytometry - Suspensions containing T cells were stained with a fixable live/dead stain (Invitrogen) in PBS followed by surface antibody staining in FACS buffer (PBS with 0.5% BSA and 0.1% sodium azide). For intracellular staining, cells were stained for surface molecules following by fixation and permeabilization (eBioscience). For cytokine staining, cells were first stimulated with Cell Stimulation Cocktail (eBioscience) containing PMA/Ionomycin and protein transport inhibitor prior to undergoing staining. For phosphostaining, 4% formaldehyde was used for fixation and 100% methanol was used for permeabilization protocols. Cells were analyzed by flow cytometry (LSRII; BD Bioscience). Data analysis was performed using FlowJo software (TreeStar).

**PP2A phosphatase assay**. In the in vivo study, mice were treated with LB-100 at 0.16 mg kg⁻¹ every other day. 4 h after the third injection, the spleens were harvested. Mouse CD3+ T cells were isolated from splenocytes with CD3 isolation kit (StemCell). Cells were then lysed in RIPA lysis buffer (Thermo Scientific) supplemented with protease inhibitors (Roche) for 30 min on ice. Cell lysates were sonicated for 10 s then centrifuged at maximum speed for 15 min. Supernatants containing 50 μg of total cellular protein were then assayed.

In the in vitro study, mouse CD4+ and CD8+ T cells were isolated from splenocytes with CD4 and CD8 isolation kit (StemCell) respectively. Cells were activated using immobilized anti-CD3 (10 μg ml⁻¹) and soluble anti-CD28 (2 μg ml⁻¹) for three hours. Cells were then similarly lysed with RIPA buffer as above and 50 μg of total cellular protein were assayed.

PP2A Phosphatase Assay Kit (Millipore) was used according to the manufacturer's instructions. Briefly, using the same amount of starting protein lysate for each condition, PP2A was immunoprecipitated using Anti-PP2A, C subunit (clone 1D6, Millipore) and Protein A agarose slurry. The slurry was then washed with TBS before a standard amount of threonine phosphopeptide, a substrate of PP2A, was added to the mixture. Phosphate was released as a product of the reaction. The absolute amount of phosphate released was quantified with malachite green solution, which was used as a measure of PP2A activity. Experiments were performed in triplicate, and the data are presented as a percentage mean of relative PP2A activity compared with control ± SEM.

**Cell-mediated cytotoxicity assay**. OT-1 CD8+ T cells were isolated from splenocytes of 6–8-week-old female OT-1 mice using mouse CD8 isolation kit (StemCell). Cells were cultured and expanded using immobilized anti-CD3 (10 μg ml⁻¹) and soluble anti-CD28 (2 μg ml⁻¹) for three days. The flow cytometry-based cell-mediated cytotoxicity assay was then performed using the LIVE/DEAD Cell-Mediated Cytotoxicity Kit (Thermo Scientific)[19]. The B16-Ova cells were used as TCs and were labeled with DiO for 20 min at 37 °C. The activated OT-1 cells were then pre-treated with LB-100 for 4 h before mixing with the TC at the specified ET ratio. The cell mixture was then incubated for 3 h in the presence of PI and analyzed by FACS. The percentage (%) of effector cell induced cell death was calculated by subtracting the % of DiO+PI+ cells in the absence of effector cells from % of DiO+PI+ cells in each condition, thereby correcting for the rate of spontaneous target cell death.

**Conjugation formation assay**. OT-1-CD8+ T cells were similarly isolated and activated as above. OT-1-CD8+ T cells were then treated with LB-100 for 5 h and then labeled with green florescent CMFDA dye (Thermo Scientific). B16-Ova TCs were labeled with orange florescent CMRA dye (Thermo Scientific). The CTLs and TC were then mixed and incubated for 1 h before analyzed by FACS. Conjugate formation was determined as % of CMFDA+CMRA+ double positive cells.

**T-cell stimulation and skewing**. Naive CD4 cells were isolated from mice splenocytes (StemCell). Cells were activated for 3 days using immobilized anti-CD3 (10 μg ml⁻¹) and soluble anti-CD28 (2 μg ml⁻¹). Skewing conditions were as follows: T$_H$1, 1 μg ml⁻¹ anti-IL4, 5 ng ml⁻¹ IL2, and 10 ng ml⁻¹ IL12; T$_H$2, 1 μg ml⁻¹ anti-IFN-γ, 5 ng ml⁻¹ IL2, and 10 ng ml⁻¹ IL4; T$_{reg}$, 1 μg ml⁻¹ anti-IFN-γ, and 1 μg ml⁻¹ anti-IL4, and 2 ng ml⁻¹ TGFβ1. Bead-based multianalyte flow immunoassays (BD Bioscience) were used to measure cytokine production in the supernatant per manufacturer's instruction. Absolute cell numbers were quantified with flow cytometry using counting beads (Biolegend).

**Antibodies for flow cytometry**. Anti-mouse: α-CD45 (30-F11, BD, 4 μg ml⁻¹), α-CD3 (145-2C11, Biolegend, 5 μg ml⁻¹), α-CD4 (GK1.5, Biolegend, 5 μg ml⁻¹), α-CD8 (53-6.7, BD, 5 μg ml⁻¹), α-CTLA-4 (1B8, abcam, 30 μg ml⁻¹), α-TIM3 (B8.2C12, Biolegend, 2.5 μg ml⁻¹), α-OX40 (OX-86, Biolegend, 10 μg ml⁻¹), α-CD62L (MEL-14, BD, 5 μg ml⁻¹), α-CD44 (IM7, Biolegend, 2.5 μg ml⁻¹), α-LAG-3 (C9B7W, Biolegend, 5 μg ml⁻¹), α-IFN-γ (XMG1.2, Biolegend, 2.5 μg ml⁻¹), α-TNF-α (MP6-XT22, Biolegend, 2.5 μg ml⁻¹), α-Granzyme B (NGZB, ThermoFisher, 1.25 μg ml⁻¹), α-FOXP3 (MF-14, Biolegend, 10 μg ml⁻¹), α-Ki67 (SolA15, ThermoFisher, 0.6 μg ml⁻¹). Anti-human: α-CD4 (A161A1,

Biolegend, 2.5 μg ml⁻¹), α-T-bet (4B10, Biolegend, 2.5 μg ml⁻¹), α-Phospho-Akt (Ser473) (D9E, Cell Signaling, 0.5 μg ml⁻¹), α-Phospho-Akt (Thr308) (D25E6, Cell Signaling, 0.5 μg ml⁻¹), α-Phospho-S6 Ribosomal Protein (Ser235/236) (D57.2.2E, Cell Signaling, 0.5 μg ml⁻¹).

**Histology**. Formalin-fixed tissues were processed, stained with hematoxylin and eosin and evaluated blindly by a board-certified pathologist.

**Human mixed lymphocyte reaction**. Dendritic cells (DCs) were generated by culturing monocytes isolated from PBMC using a monocyte isolation kit (StemCell) in vitro for 7 days with 500U ml⁻¹ interleukin-4 (IL4) and 250 U ml⁻¹ GM-CSF (R&D Systems). CD4+ T cells ($1 \times 10^5$) isolated with CD4 isolation kit (StemCell) and labeled with CFSE (ThermoFisher) were co-cultured with allogeneic DCs ($1 \times 10^4$). At the initiation of assay, a titration of LB-100 and/or Nivolumab was added. After 3 days, LB-100 was replenished to the final indicated concentration. After 5 days, culture supernatants were analyzed by ELISA (eBioscience) and cells were analyzed by flow cytometry. At least three separate donors were obtained and results of one representative donor were reported.

**Statistics**. If not stated otherwise in the figure legend, samples were analyzed with GraphPad Prism software. Scatter dot plots and bar graphs are depicted as means with SEM.

**Data availability**. The authors declare that all the other data supporting the findings of this study are available within the article and its supplementary information files and from the corresponding author upon reasonable request.

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

## Acknowledgements

The research was supported by the Intramural Research Program of the NINDS and NCI of the National Institutes of Health. We thank D. Maric, F. Livak, and K. Wolcott for expertise with flow cytometry; L. Gattinoni for providing the B16-Ova cell line; S. Walbridge for expertise with mouse handling; K. Benson and C. Butler for administrative help.

## Author contributions

W.S.H., J.S.K., J.D.H., F.M.M., M.R.G., and Z.Z. wrote the manuscript. W.S.H., H.W., Z. Z., and R.L. designed all experiments. W.S.H., H.W., R.L., Q.Z., Q.S., and D.M. carried out all experiments and data analysis.

## Additional information

**Competing interests:** J.S.K. has ownership interest in Lixte Biotechnology Holdings, Inc. A.S. The remaining authors declare no competing interests.

