## [Peer Review File · Nature Communications]

Reviewers' comments:

Reviewer #1 (Remarks to the Author):

This is a novel and well written paper on the combination of a small molecule PP2A inhibitor and PD-1 blockade for tumor immunotherapy. The combination is more effective than monotherapy and the authors analyze the immune populations in detail. The PP2A inhibitor is shown to lead to enhanced mTORC1 activity with enhancement of Th1, and decrease of Th2 and Treg. Because the levels of Treg are reduced by the PP2A inhibitor, the authors do a histologic analysis of multiple organs for autoimmune inflammation and report no therapy induced changes. This manuscript strongly supports clinical development of the combination and provides some guidance on optimal dosing.

1. Throughout the paper and abstract the authors say Treg are depleted but do not show the therapy leads to death of Treg or direct depletion. Other cells may simply be proliferating more. Consequently do not say "depleted" but rather say reduced numbers or some description directly supported by the data.
2. In 3L,K does the PD-1 FACS mab see a different epitope from the RMP1-14 treatment mab? Methods says J43 was used which is reported to be a blocker so most likely the RMP1-14 blocks the J43. Consequently none of the PD-1 staining in the treated mice is valid. Either show J43 is not blocked by RMP1-14 or remove all the PD-1 staining of PD-1 mab treated mice. Could re-do with an mab such as RMP1-30 which is non-blocking but not necessary.
3. Summarize what is known of the pharmacodynamics, half-life of PP2A inhibitor.

Minor:

4. line 42. Either name all FDA approved PD-1 and PD-L1 mabs or none
5. line 82. Text says 0.25×10^6 tumor cells but figure and methods say 0.5. Correct one is?
6. 2A-C on T cell memory, should be supplemental
7. In 4D is 4T1 labeled correctly?
8. Line 196, tumor cells are injected 2 days after start of treatment? Is this correct? Provide explanation.
9. Line 197. "Treatments were administered every 2 days after survival." Meaning unclear?
10. Line 201. Survival was prolonged. Note there were no long term survivors.
11. I calculate a 150,000 MW antibody at 0.5 nM is 75 micrograms per ml. This is high and unphysiologic. 10 ug/ml is typically used in vitro cultures. Please check the amount used and put both the nM and ug/ml amounts in the figure legend, not figure label.
12. PP2A assay, describe IP of PP2A and how samples are compared.

Reviewer #2 (Remarks to the Author):

This paper has the potentially interesting and important data, but in the present form Authors missed an opportunity to elevate their contribution to the level required by Nature Communications .

Authors provide the until now missing pharmacologic evidence for a potential feasibility of PP2a inhibitors in immunotherapies of cancer. However, some experiments are missing and Reader is not provided with background discoveries of PP2a as negative regulator to be targeted for immunoenhancement.

Criticisms and suggestions:

- 1) Better description and controls for PP2A inhibitor LB-100 are needed.

Since ALL the data are about pharmacological effects of this small molecule, there is additional

need to ensure that these effects or significant part of these effects is indeed T cell-autonomous. Instead Authors just reference earlier data about pharmacological effects on tumors.

a) Authors should provide immunological controls for T cell-autonomous mechanism of effects of this inhibitor by testing whether it will further improve the anti-tumor activities of adoptively transferred T cells with silenced Ppp2r2d. This assay should be straightforward at NIH -especially by the last Author-who was the first author in (Zhou P, et al. In vivo discovery of immunotherapy targets in the tumour microenvironment. Nature 506, 52-57 (2014).

b) What is the evidence of PP2a inhibition of T cells in vivo? Not enough that T cells CAN be inhibited in vitro as authors show.

This is also important control since authors of submitted paper did earlier performed PP2A activity assays in excised tissues in vivo and found "that at 2 hours after intraperitoneal injection of LB102, PP2A activity was diminished by 20% in the flank tumors but was unchanged in brain tissue compared to controls. However, at 4 hours after treatment, PP2A activity was markedly suppressed in both tumor and brain tissues, by approximately 40%. This inhibition was incompletely reversed at 8 and 16 hours after treatment before normalizing by 24 hours"

How this time course "cooperates" with time course of T cells penetration in the tumor?

2. Additional mechanistic assays are needed to improve the mechanistic interpretation and explanation of data to a Reader. There is also an need in much improved scholarship.

a) Authors should improve their scholarship and experimental data and provide more understanding about the mechanism of action of PP2A by performing experiments and answering questions that were posed in the very first publication that suggested the negative regulatory role of ser/thr phosphatase, i.e. PP2a in paper, entitled :
MODULATION OF CYTOLYTIC T LYMPHOCYTEFUNCTIONSBYANINHIBITOR OF SERINE/THREONINE PHOSPHATASE, OKADAICACID. Enhancement of Cytolytic T Lymphocyte-Mediated Cytotoxicity.

It was published by then intramural NIH scientists ROLF E. TAFFS at al. in 1991 in Journal of Immunology, 147,722-728 .

By using the okadaic acid (OA)- an inhibitor of both PP1 and PP2a phosphatase on CTL-TC interactions Authors emphasized that observed by them biphasic ability of ser/thr phosphatase inhibitor to enhance the Ag-specific response is unique and indicated the presence of an INHIBITORY phosphoprotein phosphatase that should be considered as a participant in the down-regulation of the cell-cell interactions between CTL and TC. Then in the same 1991 paper NIH authors identified the inhibitory phosphatase as PP2A by writing in the end of their discussion that :

"The phosphatase that is inhibited by lower concentrations of OA could be PP2A"
and " Because it is the first demonstration of the possibility to enhance cell-cell contact- dependent and Ag-specific effector functions of lymphocytes, the data reported here provide the basis for future studies of phosphoproteins important in CTL-TC interactions and for the development of new immunoenhancers".

Reader will get the much better feel of the scientific background if Authors will point out that , while data in JI 1991 were the every first demonstrations of the negative role of any protein ser/thr phosphatase and-specifically of PP2A phosphatase- in activities of CTL, the subsequent use of genetic controls did strongly confirm such role of PP2A (Penghui Zhou et al.) in thier very important Nature 2014 paper that resurrected this enzyme as the potential drug target

Since the very first publication was done in the absence of very selective inhibitors of PP2A, the authors should repeat those pretty straight-forward in vitro studies suggested here in 2b using LB-100.

Simple additional experiments:

As it is now authors of submitted MS not only missed important earlier advances , but they also did not benefit from questions posed in JI 1991 paper about how ser/thr phosphatases and specifically PP1 and PP2a ARE involved in regulation of CD8+ T cells on the level of conjugate formation and anti-tumor cytotoxicity.

Authors should revisit that paper and repeat some CTL-TC assays using their more superior and selective PP2A inhibitor in order to better describe the mechanism:

- Effects of LB-100 on CTL-TC conjugates and
- Effects of LB-100 on Direct Ag-specific cytotoxicity

The reader of Nature Communications will benefit by learning that PP2A inhibitor increases killing of tumor cells in a cytotoxicity assay, therefore localizing de-inhibition to a very important event in anti-tumor immunity.

Response to Reviewers' comments:

Reviewer #1 (Remarks to the Author):

This is a novel and well written paper on the combination of a small molecule PP2A inhibitor and PD-1 blockade for tumor immunotherapy. The combination is more effective than monotherapy and the authors analyze the immune populations in detail. The PP2A inhibitor is shown to lead to enhanced mTORC1 activity with enhancement of Th1, and decrease of Th2 and Treg. Because the levels of Treg are reduced by the PP2A inhibitor, the authors do a histologic analysis of multiple organs for autoimmune inflammation and report no therapy induced changes. This manuscript strongly supports clinical development of the combination and provides some guidance on optimal dosing.

1. Throughout the paper and abstract the authors say Treg are depleted but do not show the therapy leads to death of Treg or direct depletion. Other cells may simply be proliferating more. Consequently do not say "depleted" but rather say reduced numbers or some description directly supported by the data.

We appreciate this subtle but critical point raised by the reviewer. Indeed we have not demonstrated LB-100 with or without aPD-1 could induce greater apoptosis or cell death of Tregs. We have therefore removed all references to Treg depletion and instead used "relative reduction" of Tregs in the manuscript.

2. In 3L,K does the PD-1 FACS mab see a different epitope from the RMP1-14 treatment mab? Methods says J43 was used which is reported to be a blocker so most likely the RMP1-14 blocks the J43. Consequently none of the PD-1 staining in the treated mice is valid. Either show J43 is not blocked by RMP1-14 or remove all the PD-1 staining of PD-1 mab treated mice. Could re-do with an mab such as RMP1-30 which is non-blocking but not necessary.

This is a crucial technical detail that we have admittedly overlooked. We agree that the PD-1 staining using J43 in the TILS was likely blocked by the RMP1-14 treatment and therefore invalid. We have removed all the data using J43 staining in the manuscript.

3. Summarize what is known of the pharmacodynamics, half-life of PP2A inhibitor.

We have included the pharmacokinetics studies in rats examining tissue level of both LB-100 and its metabolite enothall, a known selective inhibitor of PP2A, in plasma, liver and brain. A detailed summary is now included in supplementary table 1 and 2.

Minor:

4. line 42. Either name all FDA approved PD-1 and PD-L1 mabs or none

We have removed names of PD-1 and PD-L1 mabs.

5. line 82. Text says 0.25×10^6 tumor cells but figure and methods say 0.5. Correct one is?

We have corrected this mistake, the correct one is 0.5.

6. 2A-C on T cell memory, should be supplemental

We agree the original Fig 2B-C is redundant with Fig 2D-E. We have therefore removed Fig. 2B-C.

7. In 4D is 4T1 labeled correctly?

We believe it is labeled correctly.

8. Line 196, tumor cells are injected 2 days after start of treatment? Is this correct? Provide explanation.

Yes, in the B16 experiment, we used a “tumor prevention” model. In our experience, B16 is a highly aggressive tumor and once the tumor formed an established nodule, the mice generally have about 1 week before reaching survival endpoint. Starting treatment at the time when tumors are established is unlikely to have therapeutic effect given the advanced stage of the tumor growth trajectory. Other studies, such as the one cited below, using B16 to examine the effect of aPD-1 treatment have employed a similar paradigm. We could have started treatment shortly (1-3 days) after inoculation as other studies have done. We have chosen our strategy in hope of maximizing the chance to see a therapeutic effect. Admittedly, it is a less realistic model and that is the reason we focused most of our data on using the CT26 tumors in which we can randomize established tumors based on similar starting volume.

Kleffel et al. Cell. 2015 Sep 10;162(6):1242-56

9. Line 197. “Treatments were administered every 2 days after survival.” Meaning unclear?

This was a typo. We meant to say: “Treatments were administered every two days until mice reached survival endpoints”

10. Line 201. Survival was prolonged. Note there were no long-term survivors.

That is correct. We did not see long-term survivors in B16 tumors. Related to (8), we think B16 is an inherently highly aggressive tumor. We saw a decrease in tumor growth as well as prolonged survival but no complete remission with combination treatment. We felt the importance of showing the B16 result is to demonstrate the synergism of LB-100 and aPD-1 in more than one tumor model. Of course we cannot expect the therapeutic effects to be the exactly the same in all tumor models, but the observed synergy between LB-100 and aPD-1 is similarly observed.

11. I calculate a 150,000 MW antibody at 0.5 nM is 75 micrograms per ml. This is high and unphysiologic. 10 ug/ml is typically used in vitro cultures. Please check the amount used and put both the nM and ug/ml amounts in the figure legend, not figure label.

To convert 0.5 nM into ug/ml: $0.5 \times 10^{-9} \text{ mol/L} \times 1.5 \times 10^5 \text{ g/mol} = 0.75 \times 10^{-4} \text{ g/L} = 75 \text{ ug/L} = 0.075 \text{ ug/ml} = 75 \text{ ng/ml}$

The mixed lymphocyte reaction experiments were based on the study (cited below) of in vitro characterization of nivolumab published by the group from Bristol Meyers. As shown in Fig 2A (attached here), 0.5nM or 75 ng/ml is a very low concentration, in which there was no change in IFN γ secretion in this MLR. We have deliberately chosen such a low dosage to test if LB-100 can increase IFN γ at such a “sub-therapeutic” doses of nivolumab. The equivalent dosage in ng/ml is now included in the figure legend.

Wang et al. *Cancer Immunol Res.* 2014 Sep;2(9):846-56

12. PP2A assay, describe IP of PP2A and how samples are compared.

A more detailed description of the PP2A assay is now included in the methods section, reproduced below:

PP2A Phosphatase Assay Kit (Millipore) was used according to the manufacturer's instructions. Briefly, using the same amount of starting protein lysate for each condition, PP2A was immunoprecipitated using Anti-PP2A, C subunit (clone 1D6, Millipore) and Protein A agarose slurry. The slurry was then washed with TBS before a standard amount of threonine phosphopeptide, a substrate of PP2A, was added to the mixture. Phosphate was released as a product of the reaction. The absolute amount of phosphate released was quantified with malachite green solution, which was used as a measure of PP2A activity. Experiments were performed in triplicate, and the data are presented as a percentage mean of relative PP2A activity compared with control \pm SEM.

Reviewer #2: (Remarks to the Author):

This paper has the potentially interesting and important data, but in the present form Authors missed an opportunity to elevate their contribution to the level required by Nature Communications .

Authors provide the until now missing pharmacologic evidence for a potential feasibility of PP2a inhibitors in immunotherapies of cancer. However, some experiments are missing and Reader is not provided with background discoveries of PP2a as negative regulator to be targeted for immunoenhancement.

Criticisms and suggestions:

1) Better description and controls for PP2A inhibitor LB-100 are needed.

Since ALL the data are about pharmacological effects of this small molecule, there is additional need to ensure that these effects or significant part of these effects is indeed T cell-autonomous.

Instead Authors just reference earlier data about pharmacological effects on tumors.

a) Authors should provide immunological controls for T cell-autonomous mechanism of effects of this inhibitor by

testing whether it will further improve the anti-tumor activities of adoptively transferred T cells with silenced Ppp2r2d. This assay should be straightforward at NIH -especially by the last Author-who was the first author in (Zhou P, et al. In vivo discovery of immunotherapy targets in the tumour microenvironment. Nature 506, 52-57 (2014).

We completely agree that given PP2A is an ubiquitously expressed phosphatase and that LB-100 has previously been shown to have a chemo- and radio-sensitizing effect on tumors, it is critical to establish if the observed synergistic effect of LB-100 and aPD-1 is a "T cell-autonomous" process and not a direct effect on tumor cells. We felt we have partially addressed this point with the CD8 depletion experiments (Fig. 1F – of revised manuscript), in which we have shown the effect of combination (with complete regression of tumor) was completely abolished when CD8 cells were depleted. We could at least conclude the LB-100 + aPD-1 mediated enhanced antitumor activity is a CD8 dependent process. It is also true that we have not treated CD8 depleted mice with LB-100 alone. We therefore agree with the reviewer's point that we have not directly addressed whether LB-100 has a direct effect on tumor growth independent of its effect on the immune system. The proposed experiment, treating mice with LB-100 and adoptive transfer of PPP2R2D silenced T cells, could indeed rule out if LB-100 could confer further direct effect on tumor in addition to the effect of PP2A inhibition in T-cells. However, this model, in this cited Nature study, uses B16-Ova tumors and OT1 CD8 T cells, which is a great departure from the CT26 model we used. To similarly address this important point, we performed the experiment summarized in Fig. 1D, in which we similarly randomized mice with CT26 tumors into 4 groups (control, LB-100 alone, aPD-1 alone and combination) as in Fig. 1A, but in this case using NSG mice. NSG mice lacked T, B and NK cells and therefore we can directly ask the question if LB-100, at this given dosage, can directly impact tumor growth in the absence of adaptive immunity. The results showed that LB-100 without T cells, has no impact on the growth of tumor or survival of mice. Therefore, with this new data together with the depletion experiment, we have shown that LB-100 with/without aPD-1 has no direct effect on tumor growth in the absence of T cells and CD8 cells are essential to confer LB-100's synergistic effect with aPD-1. This should provide strong evidence that the reported effect of LB-100 is a T-cell mediated process. It is also important to re-emphasize, as we have done in the revised results and discussion, that the LB-100 dose used in this study is about one-tenth the one used in previous studies looking at its effect on tumor. It is our belief that at this low dosage the effect on tumor itself should be minimal, but as our current data show, is sufficient to elicit significant effect in T cells.

b) What is the evidence of PP2a inhibition of T cells in vivo? Not enough that T cells CAN be inhibited in vitro as authors show.

This is also important control since authors of submitted paper did earlier performed PP2A activity assays in excised tissues in vivo and found "that at 2 hours after intraperitoneal injection of LB102, PP2A activity was diminished by 20% in the flank tumors but was unchanged in brain tissue compared to controls. However, at 4 hours after treatment, PP2A activity was markedly suppressed in both tumor and brain tissues, by approximately 40%. This inhibition was incompletely reversed at 8 and 16 hours after treatment before normalizing by 24 hours"

How this time course "cooperates" with time course of T cells penetration in the tumor?

We agree that in vivo demonstration of PP2A inhibition is important, especially given the fact that we were using a significantly lower dose of LB-100 than in previous studies. We therefore, measured PP2A activity in CD3 cells isolated from spleen 4 hours after third injection of LB-100 (i.e. on day 4 after start of treatment – first injection day 0). The result is shown in Fig. 1B. We confirmed that at the given dose of 0.16 mg/kg, PP2A was inhibited by about 37% at the time point tested. We believe this result provides evidence of in vivo target engagement. We have chosen this time point because as the reviewer pointed out, in our experience from previous studies, the maximum effect on PP2A with LB-100 i.p. injection occurred around 4 hours. And given our treatment schedule was every other day injection for a prolonged period of time, we expect some "cumulative" effect of PP2A and so we chose to measure PP2A after third injection. The reviewer raised the important point regarding the pharmacokinetic of LB-100 on PP2A activity and how that correlates with T cell infiltration in tumor. Given our treatment schedule was every other day until survival endpoint, to completely characterize the pharmacokinetics of PP2A activity during treatment duration and to correlate with T cell infiltration in tumor is a very involved endeavor. While this information will be valuable in guiding design of future human clinical trials, we do not believe it will significantly add to the general conclusion of this paper.

2. Additional mechanistic assays are needed to improve the mechanistic interpretation and explanation of data to a Reader. There is also an need in much improved scholarship.

a) Authors should improve their scholarship and experimental data and provide more understanding about the mechanism of action of PP2A by performing experiments and answering questions that were posed in the very first publication that suggested

the negative regulatory role of ser/thr phosphatase, i.e. PP2a in paper, entitled :
MODULATION OF CYTOLYTIC T LYMPHOCYTEFUNCTIONSBYANINHIBITOR OF SERINE/THREONINE PHOSPHATASE, OKADAICACID. Enhancement of Cytolytic T Lymphocyte-Mediated Cytotoxicity.

It was published by then intramural NIH scientists ROLF E. TAFFS at al. in 1991 in Journal of Immunology, 147,722-728 .

By using the okadaic acid (OA)- an inhibitor of both PP1 and PP2a phosphatase on CTL-TC interactions Authors emphasized that observed by them biphasic ability of ser/thr phosphatase inhibitor to enhance the Ag-specific response is unique and indicated the presence of an INHIBITORY phosphoprotein phosphatase that should be considered as a participant in the down-regulation of the cell-cell interactions between CTL and TC. Then in the same 1991 paper NIH authors identified the inhibitory phosphatase as PP2A by writing in the end of their discussion that :

“The phosphatase that is inhibited by lower concentrations of OA could be PP2A”

and “ Because it is the first demonstration of the possibility to enhance cell-cell contact- dependent and Ag-specific effector functions of lymphocytes, the data reported here provide the basis for future studies of phosphoproteins important in CTL-TC interactions and for the development of new immunoenhancers”.

Reader will get the much better feel of the scientific background if Authors will point out that , while data in JI 1991 were the every first demonstrations of the negative role of any protein ser/thr phosphatase and-specifically of PP2A phosphatase- in activities of CTL,

the subsequent use of genetic controls did strongly confirm such role of PP2A (Penghui Zhou et al.) in thier very important Nature 2014 paper that resurrected this enzyme as the potential drug target

Since the very first publication was done in the absence of very selective inhibitors of PP2A, the authors should repeat those pretty straight-forward in vitro studies suggested here in 2b using LB-100.

Simple additional experiments:

As it is now authors of submitted MS not only missed important earlier advances , but they also did not benefit from questions posed in JI 1991 paper about how ser/thr phosphatases and specifically PP1 and PP2a ARE involved in regulation of CD8+ T cells on the level of conjugate formation and anti-tumor cytotoxicity. Authors should revisit that paper and repeat some CTL-TC assays using their more superior and selective PP2A inhibitor in order to better describe the mechanism:

- Effects of LB-100 on CTL-TC conjugates and
- Effects of LB-100 on Direct Ag-specific cytotoxicity

We are grateful to this reviewer for highlighting the Taffs et al study in 1991, which we have embarrassingly failed to discuss. It was indeed the first demonstration of PP2A as potential target that could enhance cell-cell contact dependent and Ag-specific effector functions of lymphocytes, likely through regulation of TCR mediated transmembrane signaling. This result was later corroborated by Parry et al⁵, in which they identified PP2A as the phosphatase responsible for CTLA4 mediated deactivation of Akt signaling. The most recent Zhou et al⁶ paper further highlighted the potential of PP2A as a target for immunotherapy by identifying PP2A as the most potent target to enhance antitumor immunity in an in vivo shRNA screen. We believe these 3 papers provided the scientific foundation to rationalize inhibiting PP2A for the purpose of enhancing immunotherapy by augmenting the function of cytotoxic T-cell. The Apostolidis et al. study provided further rationale for inhibiting PP2A in immunotherapy by suggesting the complementary effect of inhibiting Tregs. In our study we have decided to focus on the mechanistic sequelae suggested by this paper and its effect on mTORC1 hyper-activation and CD4 cells differentiation. The importance of our paper is to provide the pharmacologic link to translate these prior studies towards the potential clinic application. Okadaic acid is not a clinically useful compound due to its high

toxicity. We have therefore demonstrated a novel compound currently in clinical development has a potent, but previously unexplored, effect on anticancer immunity. We were remiss in not discussing the Taffs et al paper and have substantially revised the introduction and disucssion to provide a more complete scientific background of PP2A in immunity by highlighting these studies that form the scientific foundation of our study.

Parry et al. *Mol Cell Biol.* 2005 Nov;25(21):9543-53

Zhou et al. *Nature.* 2014 Feb 6;506(7486):52-7

Apostolidis et al. *Nat Immunol.* 2016 May;17(5):556-64

We also appreciate the suggestion to explore the effects of Ag specific CTL cytotoxicity and congjuation, as it would provide in vitro evidence that inhibition of PP2A has direct enhancing effect on CTL effector function. Our results are summarized in Fig. 5F-G. We used B16-Ova as target cells and activated OT1 CD8+ T cells as CTLs that recognize the cognate ova peptide. For the cell mediated cytotoxicity assay, instead of Cr release assay, we used established flow cytometry based method (Kroesen et al) simply due to familiarity with flow cytometry based methods in our lab. Similiarly, conjugation formation assay was also performed with flow cytometry.

As shown in Fig. 5F, we did see an increase in Ag specific CTL mediated cytotoxicity with LB-100 treatment as expected from the Taffs et al. study using low dose of okadaic acid. We, however, did not observe the “dualistic” response in which higher doses of drug resulted in CTL inhibition (We have tested up to 8uM, not shown). There could be serveral explanations. LB-100 is known to be a weaker inhibitor of PP2A than okadiac acid, which may explain its non-toxicity as some level of PP2A activity is essential to cell fucntion. Most of our previous studies have shown that 30-40% of PP2A inhibition resulted in biological response in the cell type of interest. It is possible that LB-100 even at the relatively high doses do not supress PP2A acitivity to the point which resulted in the observed inihibitory effect on CTL seen in Taffs et al. But it is also possible that the discrepancy is due to difference in methods and cell line. However, in our own in vitro experiment in Fig. 8 using human dendritic-T cell mixed lymphocyte reaction, we see a similar “dualisitic “ response with LB-100. Low concentration increased proliferation and IFNg secretion but high concentration resulted in inhibition. This effect was realiably obseved in many experimental trials. Therefore, we do believe this dualisitic response is a characteristic of PP2A inhibition. In fact this dualistic response has also been reported in in tumors with PP2A inhibition (Gehring et al.) It is also important to point out the difference in cell mediated cytotoxicity assay and MLR. In the case of MLR, the T cells were treated with LB-100 after mixture with dendritic cells for the 5 days of activation. LB-100 was also re-dosed at day 3 and so the T-cells were exposed to LB-100 for a significantly period of time than the 4 hours of pre-treatment in cell mediated cytotoxicity assay. This could explain why we observed the inhibitory effect of high dose treatment in MLR but not in cytotoxicity assay. Finally, consistent with Fig. 5 in Taffs et al. (attached below), we did not see a change in conjugation formation at low doses of PP2A inhibition. We similarly observed a decrease in conjugation formation at higher doses 4uM and 8 uM (data not shown). We believe we have demonstrated LB-100 could enhance Ag-specific cell mediated cytotoxicity, consistent with the observation demonstrated in the important Taffs et al. study.

Figure 8. Effect of OA on conjugate formation. CTL/DC4 were pre-activated with indicated concentrations of OA for 4 hours and tested with 2015 TC in conjugate formation assay of the 20- or 30 min incubation (■) in the presence or absence of 2 mM EDTA. □, 20-min incubation with EDTA; ■, 30-min incubation with EDTA.

Kroesen et al. J Immunol Methods. 1992 Nov 25;156

Gehring et al. FEBS Lett. 2004 Jan 16;557(1-3):1-8

Taffs et al. J Immunol. 1991 Jul 15;147(2):722-8

REVIEWERS' COMMENTS:

Reviewer #1 (Remarks to the Author):

The authors have done additional work and addressed my concerns. This is a novel and well written paper on the combination of a small molecule PP2A inhibitor and PD-1blockade for tumor immunotherapy. The combination is more effective than monotherapy and the authors analyze the immune populations in detail. The PP2A inhibitor is shown to lead to enhanced mTORC1 activity with enhancement of Th1, and decrease of Th2 and Treg. Because the levels of Treg are reduced by the PP2A inhibitor, the authors do a histologic analysis of multiple organs for autoimmune inflammation and report no therapy induced changes. This manuscript strongly supports clinical development of the combination and provides some guidance on optimal dosing. In the discussion, the authors note that there are 96 possible combinations of structural, catalytic, and regulatory subunits for the PP2A holoenzyme. If the target preference or site of action of LB-100 is known, please add this in the discussion and reference.

Reviewer #2 (Remarks to the Author):

In this revised paper Authors did address all my concerns. They performed additional experiments that strengthened their message.
They also improved their scholarship.

This is an important contribution and I expect that publication of this MS will have an impact and translational significance.

Response to Reviewers' comments:

Reviewer #1 (Remarks to the Author):

The authors have done additional work and addressed my concerns. This is a novel and well written paper on the combination of a small molecule PP2A inhibitor and PD-1 blockade for tumor immunotherapy. The combination is more effective than monotherapy and the authors analyze the immune populations in detail. The PP2A inhibitor is shown to lead to enhanced mTORC1 activity with enhancement of Th1, and decrease of Th2 and Treg. Because the levels of Treg are reduced by the PP2A inhibitor, the authors do a histologic analysis of multiple organs for autoimmune inflammation and report no therapy induced changes. This manuscript strongly supports clinical development of the combination and provides some guidance on optimal dosing. In the discussion, the authors note that there are 96 possible combinations of structural, catalytic, and regulatory subunits for the PP2A holoenzyme. If the target preference or site of action of LB-100 is known, please add this in the discussion and reference.

LB-100 is a derivative of canthardin and its chemical structure has previously been published (Lu 2009). Prior study has shown that the family of canthardin derivative inhibits the catalytic subunit of PP2A (Li 1993) but to our knowledge the crystal structure of such interaction has not been reported. We have now included this in our discussion and added the appropriate references.

Lu et al. Proc Natl Acad Sci U S A. 2009 Jul 14;106(28):11697-702

Li et al. Biochem Pharmacol. 1993 Oct 19;46(8):1435-43.

Reviewer #2 (Remarks to the Author):

In this revised paper Authors did address all my concerns. They performed additional experiments that strengthened their message. They also improved their scholarship.

This is an important contribution and I expect that publication of this MS will have an impact and translational significance.